



# A new mechanism for atmospheric mercury redox chemistry: Implications for the global mercury budget

Hannah M. Horowitz[1], Daniel J. Jacob[1,2], Yanxu Zhang[2], Theodore S. Dibble[3], Franz Slemr[4], Helen M. Amos[2], Johan A. Schmidt[2,5], Elizabeth S. Corbitt[1], Eloïse A. Marais[2], and Elsie M. Sunderland[2,6]

[1]Department of Earth & Planetary Sciences, Harvard University, Cambridge, MA, USA
[2]Harvard John A. Paulson School of Engineering and Applied Sciences, Harvard University, Cambridge, MA, USA
[3]Chemistry Department, State University of New York-Environmental Science and Forestry, Syracuse, NY, USA
[4]Max-Planck-Institute for Chemistry (MPI-C), Department of Atmospheric Chemistry, Mainz, Germany
[5]Department of Chemistry, University of Copenhagen, Universitetsparken 5, 2100 Copenhagen O, Denmark
[6]Department of Environmental Health, Harvard T. H. Chan School of Public Health, Boston, MA, USA

*Correspondence to:* Hannah M. Horowitz (hmhorow@post.harvard.edu)

**Abstract.** Mercury (Hg) is emitted to the atmosphere mainly as volatile elemental $Hg^0$. Oxidation to water-soluble $Hg^{II}$ controls Hg deposition to ecosystems. Here we implement a new mechanism for atmospheric $Hg^0/Hg^{II}$ redox chemistry in the GEOS-Chem global model and examine the implications for the global atmospheric Hg budget and deposition patterns. Our simulation includes a new coupling of GEOS-Chem to an ocean general circulation model (MITgcm), enabling a global 3-D representation of atmosphere-ocean $Hg^0/Hg^{II}$ cycling. We find that atomic bromine (Br) of marine organobromine origin is the main atmospheric $Hg^0$ oxidant, and that second-stage HgBr oxidation is mainly by the $NO_2$ and $HO_2$ radicals. The resulting lifetime of tropospheric $Hg^0$ against oxidation is 2.7 months, shorter than in previous models. Fast $Hg^{II}$ atmospheric reduction must occur in order to match the ~6-month lifetime of Hg against deposition implied by the observed atmospheric variability of total gaseous mercury (TGM ≡ $Hg^0+Hg^{II}(g)$). We implement this reduction in GEOS-Chem as photolysis of aqueous-phase $Hg^{II}$-organic complexes in aerosols and clouds, resulting in a TGM lifetime of 5.2 months against deposition and matching both mean observed TGM and its variability. Model sensitivity analysis shows that the interhemispheric gradient of TGM, previously used to infer a longer Hg lifetime against deposition, is misleading because southern hemisphere Hg mainly originates from oceanic emissions rather than transport from the northern hemisphere. The model reproduces the observed seasonal TGM variation at northern mid-latitudes (maximum in February, minimum in September) driven by chemistry and oceanic evasion, but does not reproduce the lack of seasonality observed at southern hemisphere marine sites. Aircraft observations in the lowermost stratosphere show a strong TGM-ozone relationship indicative of fast $Hg^0$ oxidation, but we show that this relationship provides only a weak test of Hg chemistry because it is also influenced by mixing. The model reproduces observed Hg wet deposition fluxes over North America, Europe, and China, including the maximum over the US Gulf Coast driven by HgBr oxidation by $NO_2$ and $HO_2$. Low Hg wet deposition observed over rural China is attributed to fast $Hg^{II}$ reduction in the presence of high organic aerosol concentrations. We find that 80% of global $Hg^{II}$ deposition takes place over the oceans, reflecting the marine origin of Br and low concentrations of marine organics for $Hg^{II}$ reduction, and most of that deposition takes place to the tropical oceans due to the availability of $HO_2$ and $NO_2$ for second-stage HgBr oxidation.





## 1. Introduction

Atmospheric mercury (Hg) cycles between two stable redox forms, elemental ($Hg^0$) and divalent ($Hg^{II}$). Most Hg emissions are as gaseous elemental $Hg^0$, which is relatively inert and sparingly soluble in water. $Hg^0$ is

oxidized in the atmosphere to $Hg^{II}$ by loss of its two $6s^2$ electrons. $Hg^{II}$ salts are water-soluble, partition into the aerosol, are efficiently removed from the atmosphere by wet and dry deposition, but can also be reduced back to $Hg^0$. Understanding atmospheric Hg redox chemistry is critical for determining where Hg will be deposited globally. Here we propose an updated $Hg^0$/$Hg^{II}$ redox mechanism, incorporating recent kinetic data and other observational constraints, for use in atmospheric models. We implement the mechanism in the GEOS-Chem global model (Bey et

al., 2001; Holmes et al., 2010) and examine the implications for the global atmospheric Hg budget and deposition patterns.

Older models assumed gas-phase OH and ozone to be the dominant $Hg^0$ oxidants (Bergan and Rodhe, 2001; Dastoor and Larocque, 2004; Selin et al., 2007; Bullock et al., 2008; Travnikov and Ilyin, 2009; De Simone et al., 2014; Gencarelli et al., 2014). It is now well established that the corresponding HgOH and HgO products are too

thermally unstable to enable oxidation to $Hg^{II}$ under atmospheric conditions (Shepler and Peterson, 2003; Goodsite et al., 2004; Calvert and Lindberg, 2005; Hynes et al., 2009; Jones et al., 2016). The importance of Br atoms for $Hg^0$ oxidation was first recognized to explain atmospheric mercury depletion events occurring in the Arctic boundary layer in spring (Schroeder et al., 1998), where sea salt photochemistry provides a large Br source (Fan and Jacob, 1992; Steffen et al., 2008; Simpson et al., 2015). Oxidation of $Hg^0$ to $Hg^{II}$ by Br is a two-stage exothermic

mechanism where the second stage conversion of $Hg^I$ to $Hg^{II}$ can be carried out by a number of radical oxidants (Goodsite et al., 2004; Dibble et al., 2012). Holmes et al. (2006) first suggested Br atoms could be the main $Hg^0$ oxidant on a global scale, with Br originating principally from photochemical decomposition of bromoform emitted by phytoplankton (Yang et al., 2005). More recent observations of background tropospheric BrO point to ubiquitous Br radical chemistry in the troposphere (Theys et al., 2011; Wang et al., 2015). Recent aircraft observations of $Hg^{II}$

and BrO over the Southeast US are consistent with Br atoms acting as the main $Hg^0$ oxidant (Gratz et al., 2015; Shah et al., 2016).

Atmospheric $Hg^{II}$ can be deposited or alternatively reduced back to $Hg^0$. Competition between these two processes determines the global patterns of Hg deposition as well as the regional fate of $Hg^{II}$ emitted directly by combustion. Atmospheric reduction mechanisms for $Hg^{II}$ are poorly understood but photoreduction in aquatic

systems has been widely observed (Costa and Liss, 1999; Amyot et al., 2000; Mason et al., 2001). Fast in-plume reduction of $Hg^{II}$ emitted by coal-fired power plants was reported by Edgerton et al. (2006) and Lohman et al. (2006) but more recent field observations do not suggest widespread importance (Deeds et al., 2013; Landis et al., 2014). The speciation of atmospheric $Hg^{II}$ is unknown (Jaffe et al., 2014; Gustin et al., 2015; Jones et al., 2016). It is generally assumed from chemical equilibrium considerations that the main $Hg^{II}$ species are $HgCl_2$ in the gas phase

and Hg-chloride complexes in the aqueous phase (Hedgecock and Pirrone, 2001; Selin et al., 2007; Holmes et al., 2009), but this has not been actually observed. Hg-chloride complexes are relatively resistant to photoreduction (Allard and Arsenie, 1991; Pehkonen and Lin, 1998). $Hg^{II}$ also strongly binds to organic ligands, including in



particular reduced sulfur complexes and carboxyl groups (Pehkonen and Lin, 1998; Haitzer et al., 2002; Ravichandran, 2004; Zheng and Hintelmann, 2009). Photoreduction of $Hg^{II}$ bound to dissolved organic carbon (DOC) and other organic matter has been widely reported in aquatic environments (Amyot et al., 1994; Xiao et al., 1995; O'Driscoll et al., 2006; Whalin and Mason, 2006) and could also possibly take place in organic aerosols. Bash

et al. (2014) found that including in-cloud aqueous photoreduction via organic acids based on the mechanism proposed by Si and Ariya (2008) improved the simulation of Hg wet deposition in a regional air quality model. Observed photochemically-driven shifts in the abundance of naturally occurring Hg isotopes in the atmosphere and precipitation support the occurrence of aqueous-phase photoreduction involving $Hg^{II}$-organic complexes in the atmosphere (Gratz et al., 2010; Sonke, 2011; Sonke et al., 2015).

The GEOS-Chem model has been widely used for global simulations of atmospheric Hg and biogeochemical cycling with a focus on interpreting observations (e.g., Strode et al., 2008; Weiss-Penzias et al., 2015; Zhang et al., 2016). The Hg chemical mechanism in the current standard version of GEOS-Chem (www.geos-chem.org) is based on Holmes et al. (2010). Our updated mechanism includes more recent kinetic data and is applied with an improved GEOS-Chem simulation of halogen chemistry (Schmidt et al., 2016). As part of this work, we also

introduce a new coupling of GEOS-Chem to a 3-D ocean model (Y. Zhang et al., 2015) to better interpret observed seasonal variations of atmospheric Hg in the context of both atmospheric chemistry and oceanic drivers of air-sea exchange (Soerensen et al., 2013).

**2. Chemical mechanism**

Table 1 lists the new chemical mechanism implemented in GEOS-Chem. Table 2 lists reactions proposed in the literature but subsequently estimated to be too slow to be of atmospheric relevance. Oxidation of $Hg^0$ to $Hg^{II}$ in the gas phase involves a two-stage process (1, 4) with competing reactions (2, 3):

$$Hg^0 + X + M \rightarrow Hg^I X + M \quad (1)$$

$$Hg^I X + M \rightarrow Hg^0 + X + M \quad (2)$$
$$Hg^I X + Y \rightarrow Hg^0 + XY \quad (3)$$
$$Hg^I X + Y + M \rightarrow Hg^{II} XY + M, \quad (4)$$

where X is the first-stage $Hg^0$ oxidant, Y is the second-stage $Hg^I$ oxidant, and M is a molecule of air (Goodsite et al.,

2004; Donohoue et al., 2006; Dibble et al., 2012). The $Hg^I$ intermediate has an overall lifetime of less than a second so local steady state can be assumed. Goodsite et al. (2004) found that HgBr is sufficiently stable for Br atoms to be an effective first-stage $Hg^0$ oxidant, and proposed Y ≡ OH and Br as effective radicals to carry out the second stage of oxidation to $Hg^{II}$. The Cl atom can also oxidize $Hg^0$ to produce HgCl (Balabanov and Peterson, 2003; Donohoue et al., 2005). Dibble et al. (2012) found that a broad range of radical oxidants could oxidize HgBr and HgCl

including Y ≡ $NO_2$ and $HO_2$, the most abundant atmospheric radicals. Other proposed first-stage gas-phase $Hg^0$ oxidants appear to be unimportant, including OH, $O_3$, I, $I_2$, $Br_2$, $Cl_2$, BrO, ClO, HCl, $HO_2$, and $NO_3$ (Table 2). $Hg^0$ is sparingly soluble in water but it has been suggested that fast aqueous-phase oxidation could take place in cloud





droplets (Munthe and McElroy, 1992; Lin and Pehkonen, 1997; Lin and Pehkonen, 1998; Whalin et al., 2007). We include these processes in our mechanism with $O_3(aq)$, $HOCl(aq)$, and $OH(aq)$ as oxidants.

Strong complexes between $Hg^{II}$ and organic acids may allow electrons to be transferred to $Hg^{II}$ during photolysis (Pehkonen and Lin, 1998). Gårdfeldt and Jonsson (2003) found that direct photolysis of Hg-oxalate

resulted in $Hg^{II}$ reduction and Si and Ariya (2008) reported the same for Hg-dicarboxylates. O'Driscoll et al. (2004) reported a linear increase in the efficiency of $Hg^{II}$ photoreduction in freshwater with increasing DOC concentrations. We assume here that $Hg^{II}$ photreduction is dependent on the local concentration of organic aerosol (OA) and on the $NO_2$ photolysis frequency ($j_{NO_2}$) taken as proxy for for the UV actinic flux, and further assume the aerosol to be aqueous only at relative humidity greater than 35%. The scaling factor α for the reaction (Table 1) is adjusted in

GEOS-Chem such that simulated total gaseous mercury (TGM ≡ $Hg^0$+$Hg^{II}$(g)) concentrations match the global mean observed at land stations.

### 3. GEOS-Chem model

#### 3.1 General description

We start from the standard version v9-02 of the GEOS-Chem Hg model (www.geos-chem.org; Amos et al., 2012; Song et al., 2015) and implement a new 2010 inventory for speciated anthropogenic emissions (Zhang et al., 2016) including contributions from commercial products (Horowitz et al., 2014). We then add original updates described below for atmospheric chemistry and atmosphere-ocean coupling. The standard v9-02 model includes cycling of $Hg^0$ and $Hg^{II}$ between the atmosphere, land, and the surface mixed-layer of the ocean (Selin et al., 2008;

Soerensen et al., 2010). The atmospheric model transports $Hg^0$ and $Hg^{II}$ as separate tracers. Gas-particle partitioning of $Hg^{II}$ is determined by local thermodynamic equilibrium that is a function of aerosol mass concentration and temperature (Amos et al., 2012).

Holmes et al. (2010) presented the last detailed global atmospheric budget analysis of Hg in GEOS-Chem and we use it as a point of comparison for this study. Their model version included $Hg^0$ oxidation by Br atoms with

25 Br concentrations specified in the troposphere and stratosphere from the p-TOMCAT and NASA Global Modeling Initiative models, respectively (Yang et al., 2005; Strahan et al., 2007). The standard v9-02 model now uses Br concentration fields from the GEOS-Chem tropospheric bromine simulation by Parrella et al. (2012), and features other model updates relative to Holmes et al. (2010) including gas-particle $Hg^{II}$ equilibrium partitioning, a corrected washout algorithm (Amos et al., 2012), and improvements to the 2-D surface ocean model (Soerensen et al., 2010).

We conduct a 3-year simulation for 2009-2011 driven by GEOS-5 assimilated meteorological data from the NASA Global Modeling and Assimilation Office (GMAO) with native horizontal resolution of 1/2° × 2/3° and 72 vertical levels extending up to the mesosphere. The GEOS-Chem simulation is conducted at 4°×5° horizontal resolution by regridding the GEOS-5 meteorological data. We use anthropogenic Hg emissions for the year 2010 from Zhang et al. (2016). The spatial distribution of soil Hg is determined following the method of Selin et al.

(2008), with updated soil respiration emissions (910 Mg Hg a$^{-1}$) for consistency with the mechanistic terrestrial model developed by Smith-Downey et al. (2010). Emissions from snow and re-emission of deposited Hg follow



Selin et al. (2008). The model is initialized with a 15-year simulation to equilibrate the stratosphere. We present model results as averages for the three simulation years (2009-2011).

### 3.2 Atmospheric chemistry

$Hg^0/Hg^{II}$ redox chemistry in the standard GEOS-Chem v9-02 model includes $Hg^0$ oxidation by Br atoms and $Hg^{II}$ in-cloud photoreduction as described by Holmes et al. (2010), with Br concentration fields from Parrella et al. (2012). Shah et al. (2016) updated that chemistry following Dibble et al. (2012) to include $2^{nd}$-stage oxidation of HgBr by $HO_2$, $NO_2$, and BrO, as well as new kinetics for HgBr dissociation. Here we further expand and update the chemistry using the mechanism described in Table 1. Br and Cl radical concentrations are taken from a new GEOS-

Chem simulation of tropospheric oxidant-aerosol chemistry by Schmidt et al. (2016). Unlike Parrella et al. (2012), this simulation does not include a bromine radical source from sea-salt debromination as recent evidence suggests that BrO concentrations in the marine boundary layer (MBL) are generally sub-ppt (Gómez Martín et al., 2013; Wang et al., 2015). Simulated concentrations of Br in the free troposphere are on the other hand a factor of two higher than Parrella et al. (2012), reflecting more extensive heterogeneous chemistry. Schmidt et al. (2016) show

that their simulation provides a better simulation of aircraft and satellite observations of tropospheric BrO. Stratospheric concentrations of Br and Cl species are from the GEOS Chemistry Climate Model (Liang et al., 2010) and Global Modeling Initiative (Considine et al., 2008; Murray et al., 2012), respectively.

Here we use the global 3-D monthly archive of oxidant and radical concentrations from Schmidt et al. (2016). We apply diurnal scaling following Holmes et al. (2010) to the monthly mean concentrations of Br, BrO, Cl,

ClO, and HOCl; a cosine function of the solar zenith angle for OH and $HO_2$; and $NO$-$NO_2$-$O_3$ photochemical equilibrium for $NO_2$. Fast $Hg^0$ oxidation in the polar spring boundary layer is simulated by specifying high BrO concentrations there when conditions for temperature, sea ice cover, sunlight, and atmospheric stability are met (Holmes et al., 2010). Monthly mean organic aerosol (OA) concentrations are archived from a separate v9-02 GEOS-Chem simulation including primary emissions from combustion and secondary production from biogenic and

anthropogenic hydrocarbons (Pye et al., 2010). Aqueous-phase concentrations of $Hg^0$, $O_3$, and HOCl are determined from the gas-phase partial pressures and Henry's law coefficients (Table 1). We estimate aqueous in-cloud OH concentrations following Jacob et al. (2005) as $[OH(aq)] = \beta\ [OH(g)]$ with $\beta = 1 \times 10^{-19}$ M cm$^3$ molecule$^{-1}$.

### 3.3 Atmosphere-ocean coupling

GEOS-Chem v9-02 uses a 2-D surface-slab ocean model with no lateral transport and with fixed subsurface ocean concentration boundary conditions (Soerensen et al., 2010). Oceanic circulation and mixing between the surface mixed layer and deeper ocean strongly impacts $Hg^0$ concentrations in the surface ocean through the supply of reducible $Hg^{II}$, driving changes in oceanic evasion (Soerensen et al., 2013; Soerensen et al., 2014). Here we present a new two-way coupling of the GEOS-Chem atmospheric Hg simulation to the MITgcm 3-D oceanic

general circulation model with embedded plankton ecology (Y. Zhang et al., 2015). First, the GEOS-Chem Hg model with the 2-D slab ocean is initialized over 15 years of repeated present-day emissions and meteorological data (this time is required to equilibrate the stratosphere). Starting from these initial conditions, we conduct the GEOS-





Chem simulation for the desired period (here three years, 2009-2011) and archive monthly surface air $Hg^0$ concentrations and total $Hg^{II}$ deposition fluxes. We then initialize the Y. Zhang et al. (2015) ocean model for 20 years with these archived surface boundary conditions to equilibrate the upper several hundred meters of the ocean. The monthly mean surface ocean $Hg^0$ concentrations from the final year of the simulation are then input to GEOS-

Chem, replacing the 2-D slab ocean model concentrations. The 15-year GEOS-Chem simulation is repeated with this new input. This process is iterated until convergence of results is achieved. We find that two iterations are sufficient.

## 4. Global budget of atmospheric mercury

### 4.1 Budget and lifetimes

Figure 1 (top panel) shows the simulated vertical and latitudinal distributions of annual zonal mean $Hg^0$ and $Hg^{II}$ mixing ratios. $Hg^0$ decreases rapidly in the stratosphere, while $Hg^{II}$ increases with altitude and dominates total Hg in the stratosphere. This vertical structure is driven by chemistry (Selin et al., 2007; Holmes et al., 2010) and is well established from observations (Murphy et al., 2006; Talbot et al., 2007; Lyman and Jaffe, 2012). The

stratosphere in the model accounts for 12% of total Hg atmospheric mass and is in a very different redox regime from the troposphere. Here we focus on tropospheric budgets, which are most relevant to Hg deposition. The stratosphere is discussed further in Section 4.4.

Figure 2 shows the global tropospheric Hg budget from our GEOS-Chem simulation, including rates of major $Hg^0/Hg^{II}$ reactions. The global tropospheric Hg reservoir is 3900 Mg, including 3500 Mg as $Hg^0$ and 400 Mg

as $Hg^{II}$, smaller than the 4500 Mg reservoir in Holmes et al. (2010). In both cases, the reservoir was adjusted through the $Hg^{II}$ photoreduction rate coefficient to match the observed annual mean TGM at long-term monitoring sites, mainly located in northern mid-latitudes (Figure 3). Holmes et al. (2010) used observations for 2000-2008 averaging $1.87 \pm 1.00$ ng m$^{-3}$ ($n = 39$ sites), whereas we use observations for 2007-2013 averaging $1.46 \pm 0.25$ ng m$^{-3}$ ($n = 37$). Observed atmospheric concentrations in North America and Europe have declined on average by 1-2% a$^{-1}$

since 1990, which has been attributed to: (1) phase-out of Hg from commercial products, (2) declines in Hg emissions from coal combustion as a co-benefit of $SO_2$ and $NO_x$ emission controls, (3) updated artisanal and small-scale gold mining emissions (Zhang et al., 2016). This explains some of the difference between observed concentrations in the two time periods. Observations used in Holmes et al. (2010) also included very high concentrations at four East Asian sites (3-7 ng m$^{-3}$; Nguyen et al., 2007; Sakata and Asakura, 2007; Feng et al.,

2008; Wan et al., 2009), whereas the more recent observations over East Asia used here ($n = 6$) are all less than 3 ng m$^{-3}$ (Sheu et al., 2010; Fu et al., 2012a, 2012b; H. Zhang et al., 2015a). Only one site (Mt. Changbai, China) has data available during both time periods and observed TGM concentrations decreased there by a factor of two (Fu et al., 2012b). Thus, our atmospheric loadings may be more representative of more recent (ca. 2010) conditions.

We find that the lifetime of $Hg^0$ against oxidation to $Hg^{II}$ in the troposphere is 2.7 months, with Br atom-

initiated pathways contributing 97% of total oxidation. This chemical lifetime is smaller than most prior model estimates, e.g., 6 months in Holmes et al. (2010), which used lower Br concentrations. The addition of $NO_2$ and $HO_2$ as second-stage oxidants is also important in increasing the rate of $Hg^0$ conversion to $Hg^{II}$, more than compensating





for faster thermal decomposition of HgBr relative to (Holmes et al., 2010). Our results are consistent with Shah et al. (2016), who estimated an $Hg^0$ chemical lifetime of 1.2 to 2.8 months to reproduce aircraft measurements of $Hg^{II}$ over the Southeast US in summer during the NOMADSS campaign (Gratz et al., 2015). The dominance of $NO_2$ and $HO_2$ as $Hg^I$ oxidants reflects their high atmospheric abundance and is consistent with results from a box modeling study over the Pacific (Wang et al., 2014). We find Cl atoms contribute less than 1% of $Hg^0$ oxidation globally because the supply of Cl atoms (mean tropospheric mixing ratio of $2 \times 10^{-4}$ ppt) is limited by fast conversion to the stable reservoir HCl. Aqueous-phase pathways in clouds contribute 2%. Our work suggests that the dominant $Hg^{II}$ species produced in the gas phase are BrHgONO and BrHgOOH, though the actual speciation of $Hg^{II}$ in the atmosphere would likely be modified through cycling in aerosols and clouds (Hedgecock and Pirrone, 2001; Lin et al., 2006).

The vertical distribution of $Hg^0 \rightarrow Hg^{II}$ oxidation rates is shown in the bottom panel of Figure 1 in units of number density (relevant to the mass budget) and mixing ratio (relevant to transport). Only 1% of $Hg^0$ oxidation occurs in the stratosphere, consistent with previous work (Holmes et al., 2010). Most Hg oxidation by mass occurs in the extratropical free troposphere, consistent with the Br distribution (Schmidt et al., 2016), and is faster in the northern hemisphere because of higher $NO_2$ concentrations. $Hg^0$ oxidation in Holmes et al. (2010) was fastest in the Southern Ocean MBL due to high Br concentrations from sea-salt aerosol debromination. This is now thought to be unlikely because the sea salt aerosol in that region remains alkaline (Murphy et al., 1998; Alexander et al., 2005; Schmidt et al., 2016). In general, the addition of $2^{nd}$-stage oxidants $HO_2$ and $NO_2$ shifts Hg oxidation to lower latitudes relative to Holmes et al. (2010). This has implications for Hg deposition, which we discuss in Section 5. $Hg^0$ oxidation in terms of mixing ratio features a secondary maximum in the tropical upper troposphere where the Br/BrO ratio is high (Parrella et al., 2012; Fernandez et al., 2014).

We find that the shorter chemical lifetime of $Hg^0$ in our simulation relative to Holmes et al. (2010) must be balanced by faster atmospheric reduction to reproduce observed TGM concentrations. This is implemented by adjusting the photoreduction coefficient $\alpha$ in Table 1. The resulting mean lifetime of $Hg^{II}$ against reduction in the troposphere is 13 days. Shah et al. (2016) similarly found that faster $Hg^0$ oxidation in their NOMADSS simulation required faster $Hg^{II}$ reduction, with a tropospheric $Hg^{II}$ lifetime against reduction of 19 days. The lifetime of tropospheric $Hg^{II}$ against deposition is relatively long, 26 days (see Figure 2), because most $Hg^{II}$ production occurs in the free troposphere (Figure 1). We find here that the $Hg^{II}$ lifetime against reduction is shorter than against deposition, emphasizing the importance of reduction in controlling the atmospheric Hg budget. By contrast, Holmes et al. (2010) found an adjusted $Hg^{II}$ tropospheric lifetime of 50 days against reduction and 36 days against deposition, which led them to conclude that no reduction was needed if $Hg^0$ oxidation kinetics were decreased within their uncertainty. This is no longer possible with the much faster $Hg^0$ oxidation in our mechanism. We conclude that $Hg^{II}$ reduction must take place in the atmosphere. With $Hg^{II}$ reduction, the overall lifetime of tropospheric TGM against deposition in our simulation is 5.2 months, similar to the estimate of 6.1 months in Holmes et al. (2010). We discuss the consistency of this estimate with observations in the next section.

**4.2 Global distribution**





Figure 3 compares our simulation to observed 2007-2013 TGM surface concentrations. TGM includes $Hg^0$ and the gaseous component of $Hg^{II}$. Observations include annual means at 37 land sites plus ship cruises. The mean at land stations is 1.47±0.27 ng m$^{-3}$ in the observations and 1.44 ±0.25 ng m$^{-3}$ in the model, with the good agreement in the mean reflecting the adjustment of the $Hg^{II}$ photoreduction rate coefficient as explained above. The spatial

correlation coefficient between model and observations is $r = 0.57$.

Beyond this simulation of the mean, we successfully reproduce the observed standard deviation of TGM concentrations, which places an independent constraint on the atmospheric lifetime of Hg against deposition (Junge, 1974; Hamrud, 1983). This constraint had previously been expressed in terms of the interhemispheric gradient of TGM from ship cruises, leading to TGM lifetime estimates ranging from 4.4 months to 2 years (Slemr et al., 1981;

Fitzgerald et al., 1983; Lindqvist and Rodhe, 1985; Lamborg et al., 2002). However, we find the interhemispheric gradient is not a sensitive diagnostic of lifetime because atmospheric Hg in the southern hemisphere is controlled more by intrahemispheric atmosphere-ocean exchange than by transport from the northern hemisphere. For example, changing the simulation of atmosphere-ocean exchange from the slab ocean to the MITgcm (Section 3.3) increases the TGM lifetime while also increasing the interhemispheric gradient due to changed ocean emissions. We

conducted a sensitivity simulation doubling the tropospheric TGM lifetime in the model (to 10.4 months) by decreasing the rate of $Hg^0$ oxidation. This produced less than 20% of a decline in the interhemispheric gradient, but the relative standard deviation of TGM concentrations across land stations decreased by 40%. Thus the overall variability of TGM concentrations provides a more sensitive constraint on the TGM atmospheric lifetime, while the interhemispheric gradient can be misleading. The model's ability to reproduce the observed standard deviation

across land sites supports our simulated TGM atmospheric lifetime of 5.2 months.

### 4.3. Seasonality

Figure 4 compares simulated and observed seasonal cycles of TGM concentrations at land stations in the northern mid-latitudes (30° to 60° N) and southern hemisphere (see Figure 3 for site locations). Also shown are

results from a sensitivity simulation with the 2-D surface-slab ocean model (Soerensen et al., 2010). The observed seasonal cycle is successfully reproduced in the northern hemisphere. The February maximum and September minimum are driven in the model in part by $Hg^0$ oxidation (fastest in summer, slowest in winter) and in part by ocean evasion. Oceanic $Hg^0$ evasion peaks under conditions that include high wind speeds that enhance turbulent exchange and deepen the surface mixed layer, which replenishes the supply of reducible $Hg^{II}$ from the subsurface

ocean. Evasion is also sensitive to the timing of seasonal productivity blooms in the spring and summer, which lead to enhanced scavenging of $Hg^{II}$ from the mixed layer with settling particles (Mason et al., 2001; Gårdfeldt et al., 2003; Rolfus and Fitzgerald, 2004; Andersson et al., 2007; Soerensen et al., 2010; Tseng et al., 2013). Similar to Song et al. (2015), we find that the 2-D slab ocean overestimates the observed seasonality at northern mid-latitude sites (Figure 4). Atlantic and Pacific Ocean seawater $Hg^0$ measurements indicate this overestimate is a function of

simplified ocean physics (Soerensen et al., 2014) and the boundary condition for subsurface ocean concentrations in the slab ocean model. We correct it here by the coupling to the MITgcm 3-D ocean simulation.





In the southern hemisphere, Amsterdam Island in the Indian Ocean and Cape Point on the South African coast have similar monthly average concentrations and show no significant seasonal variation (Slemr et al., 2015). By contrast, the model at those sites has strong seasonality, which has been documented previously for the standard model version v9-02 (Song et al., 2015). We find a modest improvement in moving from the slab ocean simulation

to the MITgcm but the seasonal bias is still large. Capturing the seasonality of atmospheric Hg in the Arctic using GEOS-Chem required parameterization of unique sea-ice, oceanic and riverine dynamics (Fisher et al., 2012; 2013; Y. Zhang et al., 2015) and a similar analysis for the Southern Ocean region has not yet been performed.

### 4.4. Vertical distribution and the stratosphere

$Hg^0$ concentrations in the model drop rapidly above the tropopause (Figure 1), consistent with observations (Talbot et al., 2007; Slemr et al., 2009). Modeled $Hg^0$ oxidation in the lower stratosphere is almost exclusively from Br atoms. There is no significant $Hg^{II}$ reduction there because OA concentrations and relative humidity are very low. Extensive TGM observations in the lowermost stratosphere at northern extratropical latitudes are available from the CARIBIC program that collects measurements aboard commercial aircraft (Slemr et al., 2009; 2014; 2016). The

early stratospheric data were biased low and we focus on corrected data available for April 2014-January 2015 (Slemr et al., 2016). TGM in the stratosphere is expected to be mainly $Hg^0$ because $Hg^{II}$ is incorporated into aerosol (Murphy et al., 2006; Lyman and Jaffe, 2012).

Figure 5 compares the stratospheric TGM concentrations measured in CARIBIC to model values sampled along the flight tracks. The CARIBIC data also include ozone ($O_3$) and CO concentrations (Brenninkmeijer et al.,

2007). Here we use $O_3$ concentration as a chemical coordinate for depth into the stratosphere and exclude tropospheric data as diagnosed by $[O_3]/[CO] < 1.25$ mol mol$^{-1}$ (Hudman et al., 2007). We correlate the logarithm of TGM concentrations, as a measure of first-order loss, to the $O_3$ concentrations. The observations reach higher ozone than the model, indicating that they sample air with greater stratospheric influence. The model underestimates the log(TGM)-ozone slope by a factor of 2. To test whether this underestimate is due to slow $Hg^0$ oxidation, we

performed a sensitivity simulation doubling the stratospheric $Hg^0$ oxidation rate (green line in Figure 5). The model slope increases by only 32%. This suggests that the log(TGM)-ozone relationship in the lowermost stratosphere is set in part by mixing rather than solely by chemistry (Xiao et al., 2007). The model underestimate of the log(TGM)-ozone slope could reflect excessive dynamical mixing in the lower stratosphere, a well-known problem in stratospheric transport models (Schoeberl et al., 2003; Tan et al., 2004), or errors in the timescales of air transit

across the tropopause which vary on the order of months to years (Orbe et al., 2014; Ploeger and Birner, 2016).

### 5. Implications for global Hg deposition

Figure 6 compares simulated and observed annual Hg wet deposition at sites in North America, Europe, and China for 2007-2013. The model captures the spatial variability across sites in North America relatively well

($r$=0.57). Some of this variability is driven by precipitation amount, which leads to higher modeled deposition along the Northwest coast and over the North Atlantic. The maximum along the coast of the Gulf of Mexico is due to deep convection scavenging upper tropospheric air enriched in $Hg^{II}$ (Guentzel et al., 2001; Selin and Jacob, 2008; Holmes





et al., 2016). Previous GEOS-Chem simulations with Br-initiated oxidation of $Hg^0$ failed to capture this Gulf maximum because $Hg^{II}$ production favored higher latitudes (Holmes et al., 2010; Amos et al., 2012). The inclusion of $NO_2$ and $HO_2$ as second-stage oxidants in our simulation shifts $Hg^{II}$ production to lower latitudes and we find that this enables simulation of the Gulf of Mexico maximum. This solves what was previously considered a major

objection to the Br mechanism for $Hg^0$ oxidation (Holmes et al., 2010).

Over Europe, deposition is fairly uniform and low in the model and observations. This reflects decreased anthropogenic $Hg^{II}$ emissions in the region from emissions controls on coal-fired power plants (Klimont et al., 2013; Muntean et al., 2014; Zhang et al., 2016). There is no area of frequent deep convection, unlike the Gulf of Mexico for the US.

Observations of Hg wet deposition over China have recently become available (Fu et al., 2015; Fu et al., 2016). Values at urban sites are high, likely reflecting local $Hg^{II}$ emissions, and are correlated with high concentrations of particulate $Hg^{II}$ (Fu et al., 2016). Values at rural sites are much lower and comparable to wet deposition observed in North America and Europe despite higher TGM concentrations in China (see Figure 3). We simulate these low values successfully in our model, whereas a simulation with the standard GEOS-Chem v9-02

overestimates them by a factor of 4. This reflects in part our dependence of $Hg^{II}$ reduction on OA concentrations, which are particularly high in China (Heald et al., 2011).

Figure 7 shows the global distribution of wet and dry $Hg^{II}$ deposition in the model. This deposition is the main Hg source to the open ocean (Sunderland and Mason, 2007; Soerensen et al., 2010). We find that 80% of global $Hg^{II}$ deposition is to the oceans, compared to 71% in Holmes et al. (2010), and that 60% of this deposition is

wet. Because Br is of marine origin, its dominance as an $Hg^0$ oxidant favors deposition to the ocean. The dependence of $Hg^{II}$ reduction in our mechanism on the formation of Hg-organic complexes further shifts $Hg^{II}$ deposition away from continents where OA concentrations are highest (Heald et al., 2011). The dominance of $HO_2$ and $NO_2$ as second-stage oxidants for HgBr brings $Hg^{II}$ deposition to lower latitudes relative to the simulation by Holmes et al. (2010), where most deposition occurred over high-latitude oceans. We now find 49% of global total

$Hg^{II}$ deposition is to oceans within the tropics (30˚S - 30˚N). A large $Hg^{II}$ deposition flux to the tropical oceans is suggested by recent cruise observations of high oceanic $Hg^0$ concentrations across the intertropical convergence zone (ITCZ), which GEOS-Chem previously underestimated (Soerensen et al., 2014).

## 6. Conclusions

The atmospheric redox chemistry of mercury ($Hg^0$/$Hg^{II}$) determines the global patterns of Hg deposition to surface ecosystems, where Hg is converted to the toxic and bioaccumulative methylmercury species. Here we developed and evaluated an updated mechanism for atmospheric Hg redox chemistry in the GEOS-Chem global model to gain new insights into the global Hg budget and the patterns of Hg deposition. As part of this work we also developed a new coupling between GEOS-Chem and a 3-D ocean general circulation model (MITgcm), resulting in

a fully resolved simulation of Hg transport and chemistry in the atmosphere-ocean system.

The updated atmospheric $Hg^0$/$Hg^{II}$ redox mechanism includes gas-phase Br as the main $Hg^0$ oxidant in the troposphere and stratosphere, and second-stage oxidation of HgBr by a number of radical oxidants including $NO_2$



and HO$_2$. Br concentrations in GEOS-Chem are from the recent simulation of Schmidt et al. (2016), and are higher than previous models and more consistent with recent aircraft and satellite observations of BrO. Atmospheric reduction of Hg$^{II}$ is hypothesized to take place by photolysis of aqueous-phase Hg$^{II}$-organic complexes. This is parameterized in GEOS-Chem as a function of the local concentration of organic aerosol (OA).

The global mass of atmospheric Hg simulated by GEOS-Chem is 4400 Mg, including 3900 Mg in the troposphere. The tropospheric lifetime of Hg$^0$ against oxidation is 2.7 months. Observations of the atmospheric variability of total gaseous mercury (TGM ≡ Hg$^0$ + Hg$^{II}$(g)) suggest an atmospheric lifetime against deposition of about 6 months. Thus, Hg$^{II}$ must be reduced in the atmosphere, which is consistent with recent observations of atmospheric Hg isotope fractionation. Matching the observed mean surface TGM concentrations in GEOS-Chem

implies a tropospheric Hg$^{II}$ lifetime of 13 days against reduction and 26 days against deposition. This results in an overall tropospheric lifetime for TGM of 5.2 months against deposition and enables a successful simulation of the observed relative standard deviation of TGM concentrations across terrestrial sites. The interhemispheric difference in TGM concentrations had previously been interpreted to suggest a longer TGM lifetime against deposition, but we show this is misleading because TGM in the southern hemisphere is mostly controlled by oceanic emissions rather

than transport from the northern hemisphere.

The observed seasonality of TGM concentrations in northern mid-latitudes (maximum in February, minimum in September) is reproduced by the model, where it is attributed to photochemical oxidation of Hg$^0$ and oceanic evasion both with similar seasonal phases. Coupling GEOS-Chem to the MITgcm 3-D ocean model improves simulation of the seasonal amplitude by lowering oceanic evasion over the North Atlantic due to

elimination of the static subsurface ocean boundary condition in the GEOS-Chem slab ocean. Observations at southern hemisphere sites show little seasonality whereas the model features a spring maximum. More work is needed to understand this discrepancy.

Observations show a rapid depletion of TGM above the tropopause and we examined whether this could provide a test for Hg$^0$ chemistry. For that purpose we used extensive lowermost-stratospheric observations from the

CARIBIC program aboard commercial aircraft, and characterized TGM first-order loss as the slope of the log(TGM)-ozone relationship. We find that the model underestimates the observed log(TGM)-ozone slope by a factor of two, but that this slope is only moderately sensitive to the rate of Hg$^0$ oxidation and appears to be driven in part by mixing of air parcels with different stratospheric ages. This mixing is likely too fast in GEOS-Chem, which may explain the weaker TGM-ozone slope.

Hg wet deposition fluxes in the model are consistent with observations in North America, Europe, and China, lending confidence to the simulated global atmospheric Hg budget. Inclusion of NO$_2$ and HO$_2$ as second-stage HgBr oxidants in the model shifts Hg$^{II}$ production to lower latitudes compared to previous versions of GEOS-Chem and enables the model to capture the observed maximum in wet deposition along the Gulf Coast of the US. In rural areas of China, wet deposition is observed to be low in spite of very high TGM concentrations. This is

reproduced by the model where it is due to fast Hg$^{II}$ reduction, driven in part by very high OA concentrations.

We find that 80% of global Hg$^{II}$ deposition takes place over the oceans, reflecting in part the marine origin of Br as well as the relatively low marine OA concentrations and hence slow Hg$^{II}$ reduction. More Hg is deposited to





the tropical oceans (49% of total Hg$^{II}$ deposition) compared to previous versions of GEOS-Chem where debromination of sea salt aerosol drove fast Hg$^0$ oxidation and deposition to the Southern Ocean. This change largely reflects the dominance of photochemically driven Hg$^0$ oxidation in the free troposphere due to higher free tropospheric Br concentrations and the addition of the atmospherically abundant NO$_2$ and HO$_2$ radicals as second-

stage oxidants for HgBr. Observations of the latitudinal gradient of Hg$^{II}$ wet deposition over the ocean would provide a sensitive test of Hg chemistry and improve understanding of Hg inputs to different ocean regions.

**Acknowledgements.** This work was funded by the Atmospheric Chemistry and Chemical Oceanography Programs of the US National Science Foundation. We thank Oleg Travnikov for providing quality-controlled EMEP wet

deposition measurements, and the CARIBIC team (www.caribic-atmospheric.com) for maintaining this valuable measurement program. We thank Jeroen Sonke for helpful discussions.

Alexander, B., Park, R. J., Jacob, D. J., Li, Q. B., Yantosca, R. M., Savarino, J., Lee, C. C. W., and Thiemens, M. H.: Sulfate formation in sea-salt aerosols: Constraints from oxygen isotopes, Journal of Geophysical

Research-Atmospheres, 110, 10.1029/2004jd005659, 2005.

Allard, B., and Arsenie, I.: Abiotic reduction of mercury by humic substances in aquatic system - an important process for the mercury cycle, Water Air and Soil Pollution, 56, 457-464, 10.1007/bf00342291, 1991.

Amos, H. M., Jacob, D. J., Holmes, C. D., Fisher, J. A., Wang, Q., Yantosca, R. M., Corbitt, E. S., Galarneau, E., Rutter, A. P., Gustin, M. S., Steffen, A., Schauer, J. J., Graydon, J. A., St Louis, V. L., Talbot, R. W.,

Edgerton, E. S., Zhang, Y., and Sunderland, E. M.: Gas-particle partitioning of atmospheric Hg(II) and its effect on global mercury deposition, Atmospheric Chemistry and Physics, 12, 591-603, 10.5194/acp-12-591-2012, 2012.

Amyot, M., Mierle, G., Lean, D., and McQueen, D.: Sunlight-Induced Formation of Dissolved Gaseous Mercury in Lake Waters, Environmental Science & Technology, 28, 2366-2371, 10.1021/es00062a022, 1994.

Amyot, M., Lean, D. R. S., Poissant, L., and Doyon, M. R.: Distribution and transformation of elemental mercury in the St. Lawrence River and Lake Ontario, Can J Fish Aquat Sci, 57, 155-163, Doi 10.1139/Cjfas-57-S1-155, 2000.

Andersson, M. E., Gårdfeldt, K., Wängberg, I., Sprovieri, F., Pirrone, N., and Lindqvist, O.: Seasonal and daily variation of mercury evasion at coastal and off shore sites from the Mediterranean Sea, Mar Chem, 104,

214-226, 10.1016/j.marchem.2006.11.003, 2007.

Balabanov, N., and Peterson, K.: Mercury and reactive halogens: The thermochemistry of Hg+{Cl-2, Br-2, BrCl, ClO, and BrO}, Journal of Physical Chemistry A, 107, 7465-7470, 10.1021/jp035547p, 2003.

Bash, J., Carlton, A., Hutzell, W., and Bullock, O.: Regional Air Quality Model Application of the Aqueous-Phase Photo Reduction of Atmospheric Oxidized Mercury by Dicarboxylic Acids, Atmosphere, 5, 1-15,

10.3390/atmos5010001, 2014.

Bergan, T., and Rodhe, H.: Oxidation of elemental mercury in the atmosphere; Constraints imposed by global scale modelling, Journal of Atmospheric Chemistry, 40, 191-212, 10.1023/a:1011929927896, 2001.





Bey, I., Jacob, D. J., Yantosca, R. M., Logan, J. A., Field, B. D., Fiore, A. M., Li, Q. B., Liu, H. G. Y., Mickley, L. J., and Schultz, M. G.: Global modeling of tropospheric chemistry with assimilated meteorology: Model description and evaluation, Journal of Geophysical Research-Atmospheres, 106, 23073-23095, 10.1029/2001jd000807, 2001.

Brenninkmeijer, C. A. M., Crutzen, P., Boumard, F., Dauer, T., Dix, B., Ebinghaus, R., Filippi, D., Fischer, H., Franke, H., Friess, U., Heintzenberg, J., Helleis, F., Hermann, M., Kock, H. H., Koeppel, C., Lelieveld, J., Leuenberger, M., Martinsson, B. G., Miemczyk, S., Moret, H. P., Nguyen, H. N., Nyfeler, P., Oram, D., O'Sullivan, D., Penkett, S., Platt, U., Pupek, M., Ramonet, M., Randa, B., Reichelt, M., Rhee, T. S., Rohwer, J., Rosenfeld, K., Scharffe, D., Schlager, H., Schumann, U., Slemr, F., Sprung, D., Stock, P.,

Thaler, R., Valentino, F., van Velthoven, P., Waibel, A., Wandel, A., Waschitschek, K., Wiedensohler, A., Xueref-Remy, I., Zahn, A., Zech, U., and Ziereis, H.: Civil Aircraft for the regular investigation of the atmosphere based on an instrumented container: The new CARIBIC system, Atmospheric Chemistry and Physics, 7, 4953-4976, 2007.

Bullock, O. R., Atkinson, D., Braverman, T., Civerolo, K., Dastoor, A., Davignon, D., Ku, J. Y., Lohman, K.,

Myers, T. C., Park, R. J., Seigneur, C., Selin, N. E., Sistla, G., and Vijayaraghavan, K.: The North American Mercury Model Intercomparison Study (NAMMIS): Study description and model-to-model comparisons, Journal of Geophysical Research-Atmospheres, 113, 10.1029/2008jd009803, 2008.

Calvert, J. G., and Lindberg, S. E.: Mechanisms of mercury removal by O-3 and OH in the atmosphere, Atmospheric Environment, 39, 3355-3367, 10.1016/j.atmosenv.2005.01.055, 2005.

Considine, D. B., Logan, J. A., and Olsen, M. A.: Evaluation of near-tropopause ozone distributions in the Global Modeling Initiative combined stratosphere/troposphere model with ozonesonde data, Atmospheric Chemistry and Physics, 8, 2365-2385, 2008.

Costa, M., and Liss, P. S.: Photoreduction of mercury in sea water and its possible implications for Hg(0) air-sea fluxes, Mar Chem, 68, 87-95, Doi 10.1016/S0304-4203(99)00067-5, 1999.

Dastoor, A. P., and Larocque, Y.: Global circulation of atmospheric mercury: a modelling study, Atmospheric Environment, 38, 147-161, 10.1016/j.atmosenv.2003.08.037, 2004.

De Simone, F., Gencarelli, C. N., Hedgecock, I. M., and Pirrone, N.: Global atmospheric cycle of mercury: a model study on the impact of oxidation mechanisms, Environ Sci Pollut R, 21, 4110-4123, 10.1007/s11356-013-2451-x, 2014.

Deeds, D., Banic, C., Lu, J., and Daggupaty, S.: Mercury speciation in a coal-fired power plant plume: An aircraft-based study of emissions from the 3640 MW Nanticoke Generating Station, Ontario, Canada, Journal of Geophysical Research-Atmospheres, 118, 4919-4935, 10.1002/jgrd.50349, 2013.

Dibble, T. S., Zelie, M. J., and Mao, H.: Thermodynamics of reactions of ClHg and BrHg radicals with atmospherically abundant free radicals, Atmospheric Chemistry and Physics, 12, 10271-10279,

10.5194/acp-12-10271-2012, 2012.



Donohoue, D., Bauer, D., Cossairt, B., and Hynes, A.: Temperature and pressure dependent rate coefficients for the reaction of Hg with Br and the reaction of Br with Br: A pulsed laser photolysis-pulsed laser induced fluorescence study, Journal of Physical Chemistry A, 110, 6623-6632, 10.1021/jp054688j, 2006.

Donohoue, D. L., Bauer, D., and Hynes, A. J.: Temperature and pressure dependent rate coefficients for the reaction
of Hg with Cl and the reaction of Cl with Cl: A pulsed laser photolysis-pulsed laser induced fluorescence study, Journal of Physical Chemistry A, 109, 7732-7741, 10.1021/jp0513541, 2005.

Edgerton, E. S., Hartsell, B. E., and Jansen, J. J.: Mercury speciation in coal-fired power plant plumes observed at three surface sites in the southeastern US, Environmental Science & Technology, 40, 4563-4570, 10.1021/es0515607, 2006.

Fan, S.-M., and Jacob, D. J.: Surface ozone depletion in Arctic spring sustained by bromine reactions on aerosols, Nature, 359, 522-524, 1992.

Fernandez, R. P., Salawitch, R. J., Kinnison, D. E., Lamarque, J. F., and Saiz-Lopez, A.: Bromine partitioning in the tropical tropopause layer: implications for stratospheric injection, Atmospheric Chemistry and Physics, 14, 13391-13410, 10.5194/acp-14-13391-2014, 2014.

Fisher, J. A., Jacob, D. J., Soerensen, A. L., Amos, H. M., Steffen, A., and Sunderland, E. M.: Riverine source of Arctic Ocean mercury inferred from atmospheric observations, Nature Geoscience, 5, 499-504, 10.1038/ngeo1478, 2012.

Fisher, J. A., Jacob, D. J., Soerensen, A. L., Amos, H. M., Corbitt, E. S., Streets, D. G., Wang, Q. Q., Yantosca, R. M., and Sunderland, E. M.: Factors driving mercury variability in the Arctic atmosphere and ocean over the
past 30 years, Global Biogeochemical Cycles, 27, 1226-1235, 10.1002/2013GB004689, 2013.

Fitzgerald, W., Gill, G. A., and Hewitt, A. D.: Air-sea exchange of mercury, in: Trace metals in sea water, edited by: al., W. e., Plenum Publishing Company, 1983.

Fu, X. W., Feng, X., Liang, P., Deliger, Zhang, H., Ji, J., and Liu, P.: Temporal trend and sources of speciated atmospheric mercury at Waliguan GAW station, Northwestern China, Atmospheric Chemistry and Physics,
12, 1951-1964, 10.5194/acp-12-1951-2012, 2012a.

Fu, X. W., Feng, X., Shang, L. H., Wang, S. F., and Zhang, H.: Two years of measurements of atmospheric total gaseous mercury (TGM) at a remote site in Mt. Changbai area, Northeastern China, Atmospheric Chemistry and Physics, 12, 4215-4226, 10.5194/acp-12-4215-2012, 2012b.

Fu, X. W., Zhang, H., Yu, B., Wang, X., Lin, C. J., and Feng, X. B.: Observations of atmospheric mercury in China:
a critical review, Atmospheric Chemistry and Physics, 15, 9455-9476, 10.5194/acp-15-9455-2015, 2015.

Fu, X. W., Yang, X., Lang, X. F., Zhou, J., Zhang, H., Yu, B., Yan, H. Y., Lin, C. J., and Feng, X. B.: Atmospheric wet and litterfall mercury deposition at urban and rural sites in China, Atmospheric Chemistry and Physics, 16, 11547-11562, 10.5194/acp-16-11547-2016, 2016.

Gårdfeldt, K., and Jonsson, M.: Is bimolecular reduction of Hg(II) complexes possible in aqueous systems of
environmental importance, Journal of Physical Chemistry A, 107, 4478-4482, 10.1021/jp0275342, 2003.

Gårdfeldt, K., Sommar, J., Ferrara, R., Ceccarini, C., Lanzillotta, E., Munthe, J., Wängberg, I., Lindqvist, O., Pirrone, N., Sprovieri, F., Pesenti, E., and Strömberg, D.: Evasion of mercury from coastal and open waters





of the Atlantic Ocean and the Mediterranean Sea, Atmospheric Environment, 37, 78-84, 10.1016/S1352-2310(03)00238-3, 2003.

Gencarelli, C. N., De Simone, F., Hedgecock, I. M., Sprovieri, F., and Pirrone, N.: Development and application of a regional-scale atmospheric mercury model based on WRF/Chem: a Mediterranean area investigation, Environ Sci Pollut R, 21, 4095-4109, 10.1007/s11356-013-2162-3, 2014.

Gómez Martín, J. C., Mahajan, A. S., Hay, T. D., Prados-Roman, C., Ordonez, C., MacDonald, S. M., Plane, J. M. C., Sorribas, M., Gil, M., Mora, J. F. P., Reyes, M. V. A., Oram, D. E., Leedham, E., and Saiz-Lopez, A.: Iodine chemistry in the eastern Pacific marine boundary layer, Journal of Geophysical Research-Atmospheres, 118, 887-904, 10.1002/jgrd.50132, 2013.

Goodsite, M., Plane, J., and Skov, H.: A theoretical study of the oxidation of Hg-0 to HgBr2 in the troposphere, Environmental Science & Technology, 38, 1772-1776, 10.1021/es034680s, 2004.

Gratz, L., Keeler, G., Blum, J., and Sherman, L.: Isotopic Composition and Fractionation of Mercury in Great Lakes Precipitation and Ambient Air, Environmental Science & Technology, 44, 7764-7770, 10.1021/es100383w, 2010.

Gratz, L. E., Ambrose, J. L., Jaffe, D. A., Shah, V., Jaegle, L., Stutz, J., Festa, J., Spolaor, M., Tsai, C., Selin, N. E., Song, S., Zhou, X., Weinheimer, A. J., Knapp, D. J., Montzka, D. D., Flocke, F. M., Campos, T. L., Apel, E., Hornbrook, R., Blake, N. J., Hall, S., Tyndall, G. S., Reeves, M., Stechman, D., and Stell, M.: Oxidation of mercury by bromine in the subtropical Pacific free troposphere, Geophys Res Lett, 42, 10.1002/2015GL066645, 2015.

Guentzel, J. L., Landing, W. M., Gill, G. A., and Pollman, C. D.: Processes influencing rainfall deposition of mercury in Florida, Environmental Science & Technology, 35, 863-873, 10.1021/es.001523+, 2001.

Gustin, M. S., Amos, H. M., Huang, J., Miller, M. B., and Heidecorn, K.: Measuring and modeling mercury in the atmosphere: a critical review, Atmospheric Chemistry and Physics, 15, 5697-5713, 10.5194/acp-15-5697-2015, 2015.

Haitzer, M., Aiken, G. R., and Ryan, J. N.: Binding of mercury(II) to dissolved organic matter: The role of the mercury-to-DOM concentration ratio, Environmental Science & Technology, 36, 3564-3570, Doi 10.1021/Es025699i, 2002.

Hamrud, M.: Residence Time and Spatial Variability for Gases in the Atmosphere, Tellus Series B-Chemical and Physical Meteorology, 35, 295-303, 1983.

Heald, C. L., Coe, H., Jimenez, J. L., Weber, R. J., Bahreini, R., Middlebrook, A. M., Russell, L. M., Jolleys, M., Fu, T. M., Allan, J. D., Bower, K. N., Capes, G., Crosier, J., Morgan, W. T., Robinson, N. H., Williams, P. I., Cubison, M. J., DeCarlo, P. F., and Dunlea, E. J.: Exploring the vertical profile of atmospheric organic aerosol: comparing 17 aircraft field campaigns with a global model, Atmospheric Chemistry and Physics, 11, 12673-12696, 10.5194/acp-11-12673-2011, 2011.

Hedgecock, I., and Pirrone, N.: Mercury and photochemistry in the marine boundary layer-modelling studies suggest the in situ production of reactive gas phase mercury, Atmospheric Environment, 35, 3055-3062, 10.1016/s1352-2310(01)00109-1, 2001.





Holmes, C. D., Jacob, D. J., and Yang, X.: Global lifetime of elemental mercury against oxidation by atomic bromine in the free troposphere, Geophys Res Lett, 33, 5, L20808 10.1029/2006gl027176, 2006.

Holmes, C. D., Jacob, D. J., Mason, R. P., and Jaffe, D. A.: Sources and deposition of reactive gaseous mercury in the marine atmosphere, Atmospheric Environment, 43, 2278-2285, 10.1016/j.atmosenv.2009.01.051, 2009.

Holmes, C. D., Jacob, D. J., Corbitt, E. S., Mao, J., Yang, X., Talbot, R., and Slemr, F.: Global atmospheric model for mercury including oxidation by bromine atoms, Atmospheric Chemistry and Physics, 10, 12037-12057, 10.5194/acp-10-12037-2010, 2010.

Holmes, C. D., Krishnamurthy, N. P., Caffrey, J. M., Landing, W. M., Edgerton, E. S., Knapp, K. R., and Nair, U. S.: Thunderstorms Increase Mercury Wet Deposition, Environmental Science & Technology, 50, 9343-

9350, 10.1021/acs.est.6b02586, 2016.

Horowitz, H. M., Jacob, D. J., Amos, H. M., Streets, D. G., and Sunderland, E. M.: Historical Mercury Releases from Commercial Products: Global Environmental Implications, Environmental Science & Technology, 48, 10242-10250, 10.1021/es501337j, 2014.

Hudman, R. C., Jacob, D. J., Turquety, S., Leibensperger, E. M., Murray, L. T., Wu, S., Gilliland, A. B., Avery, M.,

Bertram, T. H., Brune, W., Cohen, R. C., Dibb, J. E., Flocke, F. M., Fried, A., Holloway, J., Neuman, J. A., Orville, R., Perring, A., Ren, X., Sachse, G. W., Singh, H. B., Swanson, A., and Wooldridge, P. J.: Surface and lightning sources of nitrogen oxides over the United States: Magnitudes, chemical evolution, and outflow, Journal of Geophysical Research-Atmospheres, 112, 10.1029/2006jd007912, 2007.

Hynes, A. J., Donohoue, D. L., Goodsite, M. E., and Hedgecock, I. M.: Our Current Understanding of Major

Chemical and Physical Processes Affecting Mercury Dynamics in the Atmosphere and At the Air-Water/Terrestrial Interfaces, in: Mercury Fate and Transport in the Global Atmosphere, edited by: Pirrone, N., and Mason, R., Springer Science+Business Media, LLC, 427-457, 2009.

Jacob, D., Field, B., Li, Q., Blake, D., de Gouw, J., Warneke, C., Hansel, A., Wisthaler, A., Singh, H., and Guenther, A.: Global budget of methanol: Constraints from atmospheric observations, Journal of Geophysical

Research-Atmospheres, 110, 10.1029/2004jd005172, 2005.

Jaffe, D. A., Lyman, S., Amos, H. M., Gustin, M. S., Huang, J. Y., Selin, N. E., Levin, L., ter Schure, A., Mason, R. P., Talbot, R., Rutter, A., Finley, B., Jaegle, L., Shah, V., McClure, C., Arnbrose, J., Gratz, L., Lindberg, S., Weiss-Penzias, P., Sheu, G. R., Feddersen, D., Horvat, M., Dastoor, A., Hynes, A. J., Mao, H. T., Sonke, J. E., Slemr, F., Fisher, J. A., Ebinghaus, R., Zhang, Y. X., and Edwards, G.: Progress on

Understanding Atmospheric Mercury Hampered by Uncertain Measurements, Environmental Science & Technology, 48, 7204-7206, 10.1021/es5026432, 2014.

Jiao, Y., and Dibble, T. S.: First kinetic study of the atmospherically important reactions BrHg•+ NO 2 and BrHg•+ HOO, Physical Chemistry Chemical Physics, 10.1039/c6cp06276h, 2017.

Jones, C. P., Lyman, S. N., Jaffe, D. A., Allen, T., and O'Neil, T. L.: Detection and quantification of gas-phase

oxidized mercury compounds by GC/MS, Atmos Meas Tech, 9, 2195-2205, 10.5194/amt-9-2195-2016, 2016.

Junge, C. E.: Residence Time and Variability of Tropospheric Trace Gases, Tellus, 26, 477-488, 1974.



Klimont, Z., Smith, S. J., and Cofala, J.: The last decade of global anthropogenic sulfur dioxide: 2000-2011 emissions, Environ Res Lett, 8, 10.1088/1748-9326/8/1/014003, 2013.

Lamborg, C., Fitzgerald, W., O'Donnell, J., and Torgersen, T.: A non-steady state box model of global-scale mercury biogeochemistry with interhemispheric atmospheric gradients, Abstracts of Papers of the American Chemical Society, 223, U520-U520, 2002.

Landis, M., Ryan, J., ter Schure, A., and Laudal, D.: Behavior of Mercury Emissions from a Commercial Coal-Fired Power Plant: The Relationship between Stack Speciation and Near-Field Plume Measurements, Environmental Science & Technology, 48, 13540-13548, 10.1021/es500783t, 2014.

Liang, Q., Stolarski, R., Kawa, S., Nielsen, J., Douglass, A., Rodriguez, J., Blake, D., Atlas, E., and Ott, L.: Finding the missing stratospheric Br-y: a global modeling study of CHBr3 and CH2Br2, Atmospheric Chemistry and Physics, 10, 2269-2286, 2010.

Lin, C., and Pehkonen, S.: Aqueous photoreduction of divalent mercury with organic acids: Implications of mercury chemistry in the atmosphere, Abstracts of Papers of the American Chemical Society, 213, 41-ENVR, 1997.

Lin, C., and Pehkonen, S.: Oxidation of elemental mercury by aqueous chlorine (HOCl/OCl-): Implications for tropospheric mercury chemistry, Journal of Geophysical Research-Atmospheres, 103, 28093-28102, 10.1029/98jd02304, 1998.

Lin, C. J., Pongprueksa, P., Lindberg, S. E., Pehkonen, S. O., Byun, D., and Jang, C.: Scientific uncertainties in atmospheric mercury models I: Model science evaluation, Atmospheric Environment, 40, 2911-2928, 10.1016/j.atmosenv.2006.01.009, 2006.

Lindqvist, O., and Rodhe, H.: Atmospheric mercury - a review, Tellus Series B-Chemical and Physical Meteorology, 37, 136-159, 1985.

Lohman, K., Seigneur, C., Edgerton, E., and Jansen, J.: Modeling mercury in power plant plumes, Environmental Science & Technology, 40, 3848-3854, 10.1021/es051556v, 2006.

Lyman, S., and Jaffe, D.: Formation and fate of oxidized mercury in the upper troposphere and lower stratosphere, Nature Geoscience, 5, 114-117, 10.1038/ngeo1353, 2012.

Mason, R. P., Lawson, N. M., and Sheu, G. R.: Mercury in the Atlantic Ocean: factors controlling air-sea exchange of mercury and its distribution in the upper waters, Deep-Sea Res Pt Ii, 48, 2829-2853, Doi 10.1016/S0967-0645(01)00020-0, 2001.

Muntean, M., Janssens-Maenhout, G., Song, S. J., Selin, N. E., Olivier, J. G. J., Guizzardi, D., Maas, R., and Dentener, F.: Trend analysis from 1970 to 2008 and model evaluation of EDGARv4 global gridded anthropogenic mercury emissions, Sci Total Environ, 494, 337-350, 10.1016/j.scitotenv.2014.06.014, 2014.

Munthe, J., and McElroy, W. J.: Some aqueous reactions of potential importance in the atmospheric chemistry of mercury, Atmospheric Environment Part a-General Topics, 26, 553-557, 10.1016/0960-1686(92)90168-k, 1992.

Murphy, D. M., Anderson, J. R., Quinn, P. K., McInnes, L. M., Brechtel, F. J., Kreidenweis, S. M., Middlebrook, A. M., Posfai, M., Thomson, D. S., and Buseck, P. R.: Influence of sea-salt on aerosol radiative properties in the Southern Ocean marine boundary layer, Nature, 392, 62-65, Doi 10.1038/32138, 1998.





Murphy, D. M., Hudson, P. K., Thomson, D. S., Sheridan, P. J., and Wilson, J. C.: Observations of mercury-containing aerosols, Environmental Science & Technology, 40, 3163-3167, 10.1021/es052385x, 2006.

Murray, L. T., Jacob, D. J., Logan, J. A., Hudman, R. C., and Koshak, W. J.: Optimized regional and interannual variability of lightning in a global chemical transport model constrained by LIS/OTD satellite data, Journal of Geophysical Research-Atmospheres, 117, 10.1029/2012jd017934, 2012.

O'Driscoll, N., Siciliano, S., Lean, D., and Amyot, M.: Gross photoreduction kinetics of mercury in temperate freshwater lakes and rivers: Application to a general model of DGM dynamics, Environmental Science & Technology, 40, 837-843, 10.1021/es051062y, 2006.

O'Driscoll, N. J., Lean, D. R. S., Loseto, L. L., Carignan, R., and Siciliano, S. D.: Effect of dissolved organic carbon on the photoproduction of dissolved gaseous mercury in lakes: Potential impacts of forestry, Environmental Science & Technology, 38, 2664-2672, Doi 10.1021/Es034702a, 2004.

Orbe, C., Holzer, M., Polvani, L. M., Waugh, D. W., Li, F., Oman, L. D., and Newman, P. A.: Seasonal ventilation of the stratosphere: Robust diagnostics from one-way flux distributions, Journal of Geophysical Research-Atmospheres, 119, 293-306, 10.1002/2013JD020213, 2014.

Parrella, J. P., Jacob, D. J., Liang, Q., Zhang, Y., Mickley, L. J., Miller, B., Evans, M. J., Yang, X., Pyle, J. A., Theys, N., and Van Roozendael, M.: Tropospheric bromine chemistry: implications for present and pre-industrial ozone and mercury, Atmospheric Chemistry and Physics, 12, 6723-6740, 10.5194/acp-12-6723-2012, 2012.

Pehkonen, S. O., and Lin, C. J.: Aqueous photochemistry of mercury with organic acids, Journal of the Air & Waste Management Association, 48, 144-150, 1998.

Ploeger, F., and Birner, T.: Seasonal and inter-annual variability of lower stratospheric age of air spectra, Atmospheric Chemistry and Physics, 16, 10195-10213, 10.5194/acp-16-10195-2016, 2016.

Pye, H., Chan, A., Barkley, M., and Seinfeld, J.: Global modeling of organic aerosol: the importance of reactive nitrogen (NOx and NO3), Atmospheric Chemistry and Physics, 10, 11261-11276, 10.5194/acp-10-11261-2010, 2010.

Ravichandran, M.: Interactions between mercury and dissolved organic matter - a review, Chemosphere, 55, 319-331, 10.1016/j.chemosphere.2003.11.011, 2004.

Rolfus, K. R., and Fitzgerald, W. F.: Mechanisms and temporal variability of dissolved gaseous mercury production in coastal seawater, Mar Chem, 90, 126-136, 2004.

Schmidt, J. A., Jacob, D. J., Horowitz, H. M., Hu, L., Sherwen, T., Evans, M. J., Liang, Q., Suleiman, R. M., Oram, D. E., Le Breton, M., Percival, C. J., Wang, S., and Volkamer, R.: Modeling the observed tropospheric BrO background: Importance of multiphase chemistry and implications for ozone, OH, and mercury, Journal of Geophysical Research-Atmospheres, 121, doi:10.1002/2015JD024229, 2016.

Schoeberl, M. R., Douglass, A. R., Zhu, Z. X., and Pawson, S.: A comparison of the lower stratospheric age spectra derived from a general circulation model and two data assimilation systems, Journal of Geophysical Research-Atmospheres, 108, 10.1029/2002jd002652, 2003.





Schroeder, W. H., Anlauf, K. G., Barrie, L. A., Lu, J. Y., Steffen, A., Schneeberger, D. R., and Berg, T.: Arctic springtime depletion of mercury, Nature, 394, 331-332, Doi 10.1038/28530, 1998.

Selin, N. E., Jacob, D. J., Park, R. J., Yantosca, R. M., Strode, S., Jaegle, L., and Jaffe, D.: Chemical cycling and deposition of atmospheric mercury: Global constraints from observations, Journal of Geophysical Research-Atmospheres, 112, 10.1029/2006jd007450, 2007.

Selin, N. E., and Jacob, D. J.: Seasonal and spatial patterns of mercury wet deposition in the United States: Constraints on the contribution from North American anthropogenic sources, Atmospheric Environment, 42, 5193-5204, 10.1016/j.atmosenv.2008.02.069, 2008.

Selin, N. E., Jacob, D. J., Yantosca, R. M., Strode, S., Jaegle, L., and Sunderland, E. M.: Global 3-D land-ocean-atmosphere model for mercury: Present-day versus preindustrial cycles and anthropogenic enrichment factors for deposition, Global Biogeochemical Cycles, 22, 10.1029/2007gb003040, 2008.

Shah, V., Jaegle, L., Gratz, L. E., Ambrose, J. L., Jaffe, D. A., Selin, N. E., Song, S., Campos, T. L., Flocke, F. M., Reeves, M., Stechman, D., Stell, M., Festa, J., Stutz, J., Weinheimer, A. J., Knapp, D. J., Montzka, D. D., Tyndall, G. S., Apel, E. C., Hornbrook, R. S., Hills, A. J., Riemer, D. D., Blake, N. J., Cantrell, C. A., and Mauldin, R. L.: Origin of oxidized mercury in the summertime free troposphere over the southeastern US, Atmospheric Chemistry and Physics, 16, 1511-1530, 10.5194/acp-16-1511-2016, 2016.

Shepler, B. C., and Peterson, K. A.: Mercury monoxide: A systematic investigation of its ground electronic state, Journal of Physical Chemistry A, 107, 1783-1787, 10.1021/jp027512f, 2003.

Sheu, G. R., Lin, N. H., Wang, J. L., Lee, C. T., Yang, C. F. O., and Wang, S. H.: Temporal distribution and potential sources of atmospheric mercury measured at a high-elevation background station in Taiwan, Atmospheric Environment, 44, 2393-2400, 10.1016/j.atmosenv.2010.04.009, 2010.

Si, L., and Ariya, P. A.: Reduction of oxidized mercury species by dicarboxylic acids (C(2)-C(4)): Kinetic and product studies, Environmental Science & Technology, 42, 5150-5155, 10.1021/es800552z, 2008.

Simpson, W. R., Brown, S. S., Saiz-Lopez, A., Thornton, J. A., and von Glasow, R.: Tropospheric Halogen Chemistry: Sources, Cycling, and Impacts, Chem Rev, 115, 4035-4062, 10.1021/cr5006638, 2015.

Slemr, F., Seiler, W., and Schuster, G.: Latitudinal distribution of mercury over the Atlantic-Ocean, Journal of Geophysical Research-Oceans and Atmospheres, 86, 1159-1166, 10.1029/JC086iC02p01159, 1981.

Slemr, F., Ebinghaus, R., Brenninkmeijer, C., Hermann, M., Kock, H., Martinsson, B., Schuck, T., Sprung, D., van Velthoven, P., Zahn, A., and Ziereis, H.: Gaseous mercury distribution in the upper troposphere and lower stratosphere observed onboard the CARIBIC passenger aircraft, Atmospheric Chemistry and Physics, 9, 1957-1969, 2009.

Slemr, F., Weigelt, A., Ebinghaus, R., Brenninkmeijer, C., Baker, A., Schuck, T., Rauthe-Schoch, A., Riede, H., Leedham, E., Hermann, M., van Velthoven, P., Oram, D., O'Sullivan, D., Dyroff, C., Zahn, A., and Ziereis, H.: Mercury Plumes in the Global Upper Troposphere Observed during Flights with the CARIBIC Observatory from May 2005 until June 2013, Atmosphere, 5, 342-369, 10.3390/atmos5020342, 2014.

Slemr, F., Angot, H., Dommergue, A., Magand, O., Barret, M., Weigelt, A., Ebinghaus, R., Brunke, E. G., Pfaffhuber, K. A., Edwards, G., Howard, D., Powell, J., Keywood, M., and Wang, F.: Comparison of





mercury concentrations measured at several sites in the Southern Hemisphere, Atmospheric Chemistry and Physics, 15, 3125-3133, 10.5194/acp-15-3125-2015, 2015.

Slemr, F., Weigelt, A., Ebinghaus, R., Kock, H. H., Bodewadt, J., Brenninkmeijer, C. A. M., Rauthe-Schoch, A., Weber, S., Hermann, M., Becker, J., Zahn, A., and Martinsson, B.: Atmospheric mercury measurements onboard the CARIBIC passenger aircraft, Atmos Meas Tech, 9, 2291-2302, 10.5194/amt-9-2291-2016, 2016.

Smith-Downey, N. V., Sunderland, E. M., and Jacob, D. J.: Anthropogenic impacts on global storage and emissions of mercury from terrestrial soils: Insights from a new global model, Journal of Geophysical Research-Biogeosciences, 115, 10.1029/2009jg001124, 2010.

Soerensen, A. L., Sunderland, E. M., Holmes, C. D., Jacob, D. J., Yantosca, R. M., Skov, H., Christensen, J. H., Strode, S. A., and Mason, R. P.: An Improved Global Model for Air-Sea Exchange of Mercury: High Concentrations over the North Atlantic, Environmental Science & Technology, 44, 8574-8580, Doi 10.1021/Es102032g, 2010.

Soerensen, A. L., Mason, R. P., Balcom, P. H., and Sunderland, E. M.: Drivers of Surface Ocean Mercury Concentrations and Air-Sea Exchange in the West Atlantic Ocean, Environmental Science & Technology, 47, 7757-7765, 10.1021/es401354q, 2013.

Soerensen, A. L., Mason, R. P., Balcom, P. H., Jacob, D. J., Zhang, Y. X., Kuss, J., and Sunderland, E. M.: Elemental Mercury Concentrations and Fluxes in the Tropical Atmosphere and Ocean, Environmental Science & Technology, 48, 11312-11319, 10.1021/es503109p, 2014.

Song, S., Selin, N., Soerensen, A., Angot, H., Artz, R., Brooks, S., Brunke, E., Conley, G., Dommergue, A., Ebinghaus, R., Holsen, T., Jaffe, D., Kang, S., Kelley, P., Luke, W., Magand, O., Marumoto, K., Pfaffhuber, K., Ren, X., Sheu, G., Slemr, F., Warneke, T., Weigelt, A., Weiss-Penzias, P., Wip, D., and Zhang, Q.: Top-down constraints on atmospheric mercury emissions and implications for global biogeochemical cycling, Atmospheric Chemistry and Physics, 15, 7103-7125, 10.5194/acp-15-7103-2015, 2015.

Sonke, J.: A global model of mass independent mercury stable isotope fractionation, Geochimica Et Cosmochimica Acta, 75, 4577-4590, 10.1016/j.gca.2011.05.027, 2011.

Sonke, J. E., Maruszczak, N., and Fu, X. W.: A stable isotope view of the atmospheric Hg cycle, Goldschmidt, Prague, Czech Republic, August 2015, 2015.

Steffen, A., Douglas, T., Amyot, M., Ariya, P., Aspmo, K., Berg, T., Bottenheim, J., Brooks, S., Cobbett, F., Dastoor, A., Dommergue, A., Ebinghaus, R., Ferrari, C., Gardfeldt, K., Goodsite, M. E., Lean, D., Poulain, A. J., Scherz, C., Skov, H., Sommar, J., and Temme, C.: A synthesis of atmospheric mercury depletion event chemistry in the atmosphere and snow, Atmospheric Chemistry and Physics, 8, 1445-1482, 2008.

Strahan, S. E., Duncan, B. N., and Hoor, P.: Observationally derived transport diagnostics for the lowermost stratosphere and their application to the GMI chemistry and transport model, Atmospheric Chemistry and Physics, 7, 2435-2445, 2007.




Strode, S. A., Jaegle, L., Jaffe, D. A., Swartzendruber, P. C., Selin, N. E., Holmes, C., and Yantosca, R. M.: Trans-Pacific transport of mercury, Journal of Geophysical Research-Atmospheres, 113, 10.1029/2007jd009428, 2008.

Sunderland, E. M., and Mason, R. P.: Human impacts on open ocean mercury concentrations, Global Biogeochemical Cycles, 21, 10.1029/2006gb002876, 2007.

Talbot, R., Mao, H., Scheuer, E., Dibb, J., and Avery, M.: Total depletion of Hg˚ in the upper troposphere-lower stratosphere, Geophys Res Lett, 34, 10.1029/2007gl031366, 2007.

Tan, W. W., Geller, M. A., Pawson, S., and da Silva, A.: A case study of excessive subtropical transport in the stratosphere of a data assimilation system, Journal of Geophysical Research-Atmospheres, 109, 10.1029/2003jd004057, 2004.

Theys, N., Van Roozendael, M., Hendrick, F., Yang, X., De Smedt, I., Richter, A., Begoin, M., Errera, Q., Johnston, P. V., Kreher, K., and De Maziere, M.: Global observations of tropospheric BrO columns using GOME-2 satellite data, Atmospheric Chemistry and Physics, 11, 1791-1811, 10.5194/acp-11-1791-2011, 2011.

Travnikov, O., and Ilyin, I.: The EMEP/MSC-E Mercury Modeling System, Mercury Fate and Transport in the Global Atmosphere, 571-587, 10.1007/978-0-387-93958-2_20, 2009.

Tseng, C. M., Lamborg, C. H., and Hsu, S. C.: A unique seasonal pattern in dissolved elemental mercury in the South China Sea, a tropical and monsoon-dominated marginal sea, Geophys Res Lett, 40, 167-172, 10.1029/2012GL054457, 2013.

Wang, F., Saiz-Lopez, A., Mahajan, A., Martin, J., Armstrong, D., Lemes, M., Hay, T., and Prados-Roman, C.: Enhanced production of oxidised mercury over the tropical Pacific Ocean: a key missing oxidation pathway, Atmospheric Chemistry and Physics, 14, 1323-1335, 10.5194/acp-14-1323-2014, 2014.

Wang, S. Y., Schmidt, J. A., Baidar, S., Coburn, S., Dix, B., Koenig, T. K., Apel, E., Bowdalo, D., Campos, T. L., Eloranta, E., Evans, M. J., DiGangi, J. P., Zondlo, M. A., Gao, R. S., Haggerty, J. A., Hall, S. R., Hornbrook, R. S., Jacob, D., Morley, B., Pierce, B., Reeves, M., Romashkin, P., ter Schure, A., and Volkamer, R.: Active and widespread halogen chemistry in the tropical and subtropical free troposphere, Proceedings of the National Academy of Sciences of the United States of America, 112, 9281-9286, 10.1073/pnas.1505142112, 2015.

Weiss-Penzias, P., Amos, H. M., Selin, N. E., Gustin, M. S., Jaffe, D. A., Obrist, D., Sheu, G. R., and Giang, A.: Use of a global model to understand speciated atmospheric mercury observations at five high-elevation sites, Atmospheric Chemistry and Physics, 15, 1161-1173, 10.5194/acp-15-1161-2015, 2015.

Whalin, L., and Mason, R.: A new method for the investigation of mercury redox chemistry in natural waters utilizing deflatable Teflon (R) bags and additions of isotopically labeled mercury, Analytica Chimica Acta, 558, 211-221, 10.1016/j.aca.2005.10.070, 2006.

Whalin, L., Kim, E. H., and Mason, R.: Factors influencing the oxidation, reduction, methylation and demethylation of mercury species in coastal waters, Mar Chem, 107, 278-294, 10.1016/j.marchem.2007.04.002, 2007.

Xiao, Y., Jacob, D. J., and Turquety, S.: Atmospheric acetylene and its relationship with CO as an indicator of air mass age, Journal of Geophysical Research, 112, D12305, 10.1029/2006JD008268, 2007.





Xiao, Z. F., Stromberg, D., and Lindqvist, O.: Influence of Humic Substances on Photolysis of Divalent Mercury in Aqueous-Solution, Water Air and Soil Pollution, 80, 789-798, Doi 10.1007/Bf01189730, 1995.

Yang, X., Cox, R., Warwick, N., Pyle, J., Carver, G., O'Connor, F., and Savage, N.: Tropospheric bromine chemistry and its impacts on ozone: A model study, Journal of Geophysical Research-Atmospheres, 110, 10.1029/2005jd006244, 2005.

Zhang, H., Fu, X. W., Lin, C. J., Wang, X., and Feng, X. B.: Observation and analysis of speciated atmospheric mercury in Shangri-La, Tibetan Plateau, China, Atmospheric Chemistry and Physics, 15, 653-665, 10.5194/acp-15-653-2015, 2015.

Zhang, Y. X., Jacob, D., Dutkiewicz, S., Amos, H., Long, M., and Sunderland, E.: Biogeochemical drivers of the fate of riverine mercury discharged to the global and Arctic oceans, Global Biogeochemical Cycles, 29, 854-864, 10.1002/2015gb005124, 2015.

Zhang, Y. X., Jacob, D. J., Horowitz, H. M., Chen, L., Amos, H. M., Krabbenhoft, D. P., Slemr, F., St Louis, V. L., and Sunderland, E. M.: Observed decrease in atmospheric mercury explained by global decline in anthropogenic emissions, Proceedings of the National Academy of Sciences of the United States of America, 113, 526-531, 10.1073/pnas.1516312113, 2016.

Zheng, W., and Hintelmann, H.: Mercury isotope fractionation during photoreduction in natural water is controlled by its Hg/DOC ratio, Geochimica Et Cosmochimica Acta, 73, 6704-6715, 10.1016/j.gca.2009.08.016, 2009.



**Table 1. Mechanism for atmospheric mercury redox chemistry.**

| Reaction | Rate expression[a] | Reference[b] |
|---|---|---|
| *Gas phase* | | |
| $Hg^0 + Br + M \rightarrow HgBr + M$ | $1.46 \times 10^{-32} (T/298)^{-1.86} [Hg^0][Br][M]$ | (1) |
| $HgBr + M \rightarrow Hg^0 + Br + M$ | $1.6 \times 10^{-9} (T/298)^{-1.86} \exp(-7801/T)[HgBr][M]$ | (2) |
| $HgBr + Br \rightarrow Hg^0 + Br_2$ | $3.9 \times 10^{-11} [HgBr][Br]$ | (3) |
| $HgBr + NO_2 \rightarrow Hg^0 + BrNO_2$ | $3.4 \times 10^{-12} \exp(391/T)[HgBr][NO_2]$ | (4) |
| $HgBr + Br \xrightarrow{M} HgBr_2$ | $3.0 \times 10^{-11} [HgBr][Br]$ | (3)[c] |
| $HgBr + NO_2 \xrightarrow{M} HgBrNO_2$ | $k_{NO_2}([M], T) [HgBr][NO_2]$ | (4)[d] |
| $HgBr + Y \xrightarrow{M} HgBrY$ | $k_{HO_2}([M], T) [HgBr][Y]$ | (4)[d, e] |
| $Hg^0 + Cl + M \rightarrow HgCl + M$ | $2.2 \times 10^{-32} \exp(680(1/T - 1/298))[Hg^0][Cl][M]$ | (5) |
| $HgCl + Cl \rightarrow Hg^0 + Cl_2$ | $1.20 \times 10^{-11} \exp(-5942/T)[HgCl][Cl]$ | (6)[f] |
| $HgCl + Br \xrightarrow{M} HgBrCl$ | $3.0 \times 10^{-11} [HgCl][Br]$ | (3)[c, g] |
| $HgCl + NO_2 \xrightarrow{M} HgClNO_2$ | $k_{NO_2}([M], T) [HgBr][NO_2]$ | (4)[d, g] |
| $HgCl + Y \xrightarrow{M} HgClY$ | $k_{HO_2}([M], T) [HgBr][Y]$ | (4)[d, e, g] |
| *Aqueous phase*[h] | | |
| $Hg^0_{(aq)} + O_{3(aq)} \rightarrow Hg^{II}_{(aq)} + products$ | $4.7 \times 10^7 [Hg^0_{(aq)}][O_{3(aq)}]$ | (7) |
| $Hg^0_{(aq)} + HOCl_{(aq)} \rightarrow Hg^{II}_{(aq)} + OH^-_{(aq)} + Cl^-_{(aq)}$ | $2 \times 10^6 [Hg^0_{(aq)}][HOCl_{(aq)}]$ | (8), (9)[i] |
| $Hg^0_{(aq)} + OH_{(aq)} \rightarrow Hg^{II}_{(aq)} + products$ | $2.0 \times 10^9 [Hg^0_{(aq)}][OH_{(aq)}]$ | (10), (11) |
| $Hg^{II}_{(aq)} + h\nu \rightarrow Hg^0_{(aq)}$ | $\alpha j_{NO_2}[OA][Hg^{II}_{(aq)}]$ | this work[j] |

[a] Rate expressions for gas-phase reactions have units of molecule $cm^{-3} s^{-1}$ where [ ] denotes concentration in number density units of molecules $cm^{-3}$, [M] is the number density of air, and $T$ is temperature in K. Rate expressions for aqueous-phase reactions have units of $M s^{-1}$ and $[_{(aq)}]$ denotes concentration in M (mol $L^{-1}$). Henry's law coefficients (M $atm^{-1}$) relating gas-phase and aqueous-phase concentrations are $1.28 \times 10^{-1} \exp(2482(1/T - 1/298))$ for $Hg^0$ (Sanemasa, 1975), $1.1 \times 10^{-2} \exp(2400(1/T - 1/298))$ for $O_3$ (Jacob, 1986), $6.6 \times 10^2 \exp(5900(1/T - 1/298))$ for HOCl (Huthwelker et al., 1995), and $1.4 \times 10^6$ for $Hg^{II}$ ($HgCl_2$; Lindqvist and Rodhe, 1985). The concentration of $OH_{(aq)}$ is estimated as given in the text. Lifetimes of $Hg^I$ species are sufficiently short that steady-state can be assumed for atmospheric purposes.

[b] (1) Donohoue et al., 2006; (2) Dibble et al., 2012; (3) Balabanov et al., 2005; (4) Jiao and Dibble, 2017; (5) Donohoue et al. 2005; (6) Wilcox, 2009; (7) Munthe, 1992; (8) Lin and Pehkonen, 1998; (9) Wang and Pehkonen (2004); (10) Lin and Pehkonen, 1997; (11) Buxton et al., 1988.

[c] This is an effective rate constant for intermediate pressures most relevant for atmospheric oxidation and uses a higher level of theory than Goodsite et al. (2004).

[d] $k([M], T) = \left(\frac{k^0(T)[M]}{1 + k^0(T)[M]/k^\infty(T)}\right) 0.6^p$, where $p = \left(1 + \left(\log_{10}\left(k^0(T)[M]/k^\infty(T)\right)\right)^2\right)^{-1}$. Values of $k^0(T)$ and $k^\infty(T)$ are tabulated by Jiao and Dibble (2017) for different temperatures. At $T = \{220, 260, 280, 298, 320\}$ K $k^0_{NO_2} = \{27.4, 13.5, 9.52, 7.10, 5.09\} \times 10^{-29}$ $cm^6$ $molecule^{-2} s^{-1}$; $k^\infty_{NO_2} = \{22.0, 14.2, 12.8, 11.8, 10.9\} \times 10^{-11}$ $cm^3$ $molecule^{-1} s^{-1}$;





$k^0_{HO_2}$ = {8.40, 4.28, 3.01, 2.27, 1.64} × $10^{-29}$ cm$^6$ molecule$^{-2}$ s$^{-1}$; $k^\infty_{HO_2}$ = {14.8, 9.10, 7.55, 6.99, 6.11} × $10^{-11}$ cm$^3$ molecule$^{-1}$ s$^{-1}$.

[e] We assume that the rate coefficient determined for HgBr + HO$_2$ holds more generally for Y ≡ HO$_2$, OH, Cl, BrO, and ClO based on similar binding energies (Goodsite et al., 2004; Dibble et al., 2012) and *ab initio* calculations from Wang et al. (2014).

[f] Abstraction alone competes with oxidation to Hg$^{II}$ for HgCl as the rate of HgCl thermal dissociation is negligibly slow (Holmes et al., 2009).

[g] HgCl + Y rate coefficients are assumed to be the same as HgBr + Y. ClHg-Y and BrHg-Y have similar bond energies (Dibble et al., 2012).

[h] Oxidation of Hg$^0_{(aq)}$ takes place in clouds only, with concentrations of Hg$^0_{(aq)}$ and aqueous-phase oxidants determined by Henry's law equilibrium (footnote *a*). Aerosol liquid water contents under non-cloud conditions are too low for these reactions to be significant. Photoreduction of Hg$^{II}_{(aq)}$ takes place in both aqueous aerosols and clouds, with gas-aerosol partitioning of Hg$^{II}$ as given by Amos et al. (2012) outside of clouds and Henry's law equilibrium for HgCl$_2$ in cloud (footnote *a*). The aerosol is assumed aqueous if RH > 35%.

[i] OCl$^-$ has similar kinetics as HOCl but is negligible for typical cloud and aerosol pH given the HOCl/OCl$^-$ pK$_a$ = 7.53 (Harris, 2002).

[j] Parameterization for photoreduction of aqueous-phase Hg$^{II}$-organic complexes. Here $j_{NO2}$ (s$^{-1}$) is the local photolysis rate constant for NO$_2$ intended to be representative of the near-UV actinic flux, [OA] (μg m$^{-3}$ STP) is the mass concentration of organic aerosol under standard conditions of temperature and pressure ($p$ = 1 atm, $T$ = 273 K), and α = 5.2 × $10^{-2}$ m$^3$ STP μg$^{-1}$ is a coefficient adjusted in GEOS-Chem to match observed TGM concentrations (see text).





**Table 2. Reactions not included in chemical mechanism.[a]**

| Reaction | Reference[b] | Note |
|---|---|---|
| $Hg^0 + Br_2 \xrightarrow{M} HgBr_2$ | Balabanov et al. (2005); Auzmendi-Murua et al. (2014) | c |
| $Hg^0 + Cl_2 \xrightarrow{M} HgCl_2$ | Auzmendi-Murua et al. (2014) | c |
| $Hg^0 + I_2 \xrightarrow{M} HgI_2$ | Auzmendi-Murua et al. (2014) | c |
| $Hg^0 + Br_2 \rightarrow HgBr + Br$ | Auzmendi-Murua et al. (2014) | d |
| $Hg^0 + Cl_2 \rightarrow HgCl + Cl$ | Auzmendi-Murua et al. (2014) | d |
| $Hg^0 + I_2 \rightarrow HgI + I$ | Auzmendi-Murua et al. (2014) | d |
| $Hg^0 + OH \rightarrow HgO + H$ | Hynes et al. (2009) | d |
| $Hg^0 + OH \xrightarrow{M} HgOH$ | Goodsite et al. (2004); Hynes et al. (2009) | e |
| $HgOH + O_2 \rightarrow HgO + HO_2$ | Shepler and Peterson (2003); Hynes et al. (2009) | d |
| $Hg^0 + HO_2 \xrightarrow{M} HgOOH$ | Dibble et al. (2012) | e |
| $Hg^0 + NO_3 \xrightarrow{M} HgONO_2$ | Dibble et al. (2012) | e, f |
| $Hg^0 + I \xrightarrow{M} HgI$ | Greig et al. (1970); Goodsite et al. (2004); Subir et al. (2011) | e, g |
| $Hg^0 + BrO \xrightarrow{M} HgBrO$ | Balabanov and Peterson (2003); Balabanov et al. (2005); Dibble et al. (2012, 2013) | e, h |
| $Hg^0 + ClO \xrightarrow{M} HgClO$ | Balabanov and Peterson (2003); Dibble et al. (2012, 2013) | e, h |
| $Hg^0 + HCl \rightarrow Hg^{II} + products$ | Hall and Bloom (1993); Subir et al. (2011) | g |
| $Hg^0 + O_3 \rightarrow Hg^{II} + products$ | see note | i |
| $HgBr + I \xrightarrow{M} HgBrI$ | see note | j |
| $HgBr + IO \xrightarrow{M} HgBrOI$ | see note | j |
| $Hg^0_{(aq)} + HOBr_{(aq)} \rightarrow Hg^{II}_{(aq)} + OH^-_{(aq)} + Br^-_{(aq)}$ | Wang and Pehkonen (2004); Hynes et al. (2009) | k |
| $Hg^0_{(aq)} + OBr^-_{(aq)} + H^+ \rightarrow Hg^{II}_{(aq)} + OH^- + Br^-_{(aq)}$ | Wang and Pehkonen (2004); Hynes et al. (2009) | k |
| $Hg^0_{(aq)} + Br_{2(aq)} \rightarrow Hg^{II}_{(aq)} + 2Br^-_{(aq)}$ | Wang and Pehkonen (2004); Hynes et al. (2009) | k |
| $Hg^{II}_{(aq)} \xrightarrow{HO_2/O_2^-} Hg^I_{(aq)} \xrightarrow{HO_2/O_2^-} Hg^0$ | Gårdfeldt and Jonsson (2003) | d |
| $HgSO_{3(aq)} \rightarrow Hg^0_{(aq)} + products$ | van Loon et al. (2001) | l |

[a] These reactions have been inferred from kinetic studies and/or included in past models but are now thought to be too slow to be of atmospheric relevance.

[b] References supporting non-inclusion in the chemical mechanism.

[c] Reaction reported in laboratory studies by Ariya et al. (2002), Yan et al. (2005, 2009), Liu et al. (2007), Raofie et al. (2008), and Qu et al. (2010), but not supported by theory.

[d] Endothermic reaction.

[e] The $Hg^I$ compounds are weakly bound and thermally dissociate too fast to allow for 2nd step of oxidation to $Hg^{II}$.

[f] Peleg et al. (2015) find a strong correlation between observed nighttime gaseous $Hg^{II}$ and $NO_3$ radical concentrations in Jerusalem. There is theoretical evidence against $NO_3$ initiation of $Hg^0$ oxidation (Dibble et al., 2012), and $NO_3$ may instead serve as the second-stage oxidant of $Hg^I$. This would be important only in warm urban environments where nighttime $NO_3$ is high.

[g] Found to be negligibly slow.

[h] The formation of compounds with structural formulae BrHgO and ClHgO requires a prohibitively high activation energy. Note e applies to compounds with structural formulae HgBrO, HgOBr, HgClO, and HgOCl.

[i] The direct formation of $HgO_{(s)}$ from this reaction is energetically unfavorable (Calvert and Lindberg, 2005; Tossell, 2006). It has been postulated that intermediate products like $HgO_3$ could lead to the formation of stable $(HgO)_n$ oligomers or $HgO_{(s)}$ via decomposition to $OHgOO_{(g)}$ (Subir et al., 2011), but these must be stabilized through heterogeneous reactions on atmospheric aerosols (Calvert and Lindberg, 2005) and the associated mechanism is unlikely to be significant in the atmosphere (Hynes et al., 2009). A gas-phase reaction has been reported in chamber studies (Hall, 1995; Pal and Ariya, 2004; Sumner et al., 2005; Snider et al., 2008; Rutter et al., 2012), but this is





likely to have been influenced by the walls of the chamber (as seen in Pal and Ariya, 2004) and presence of
secondary organic aerosols (in Rutter et al., 2012).

[j] Atmospheric concentrations of I and IO (Dix et al., 2013; Prados-Roman et al., 2015; Volkamer et al., 2015) are
too low for these reactions to be significant.

5    [k] Rates are too slow to be of relevance.

[l] Concentrations of $HgSO_3$ under typical atmospheric $SO_2$ levels are expected to be very low.





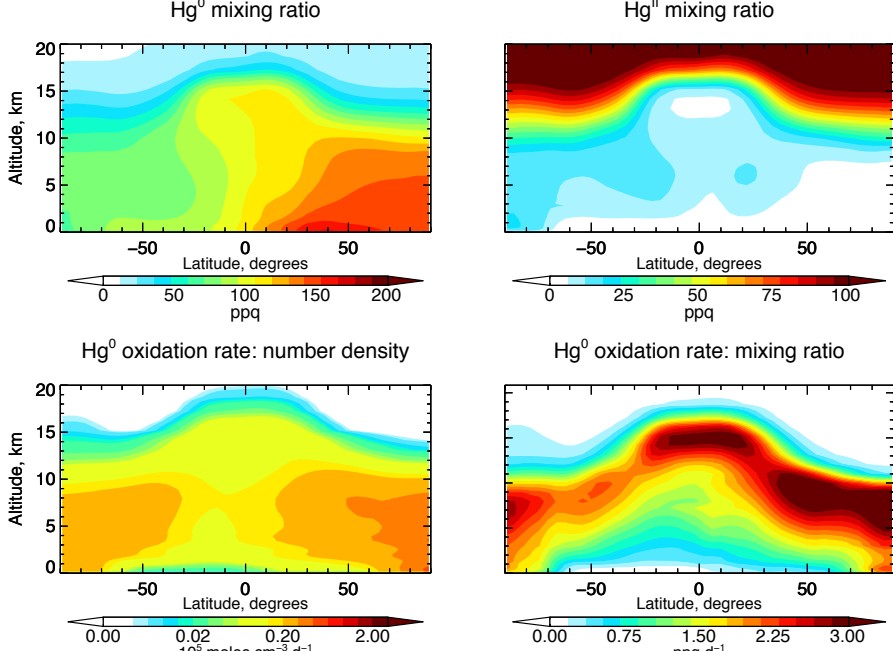

**Figure 1. Annual (2009-2011) zonal mean mixing ratios of Hg$^0$ and Hg$^{II}$ in GEOS-Chem, and Hg$^0$ oxidation rates in number density and mixing ratio units.**





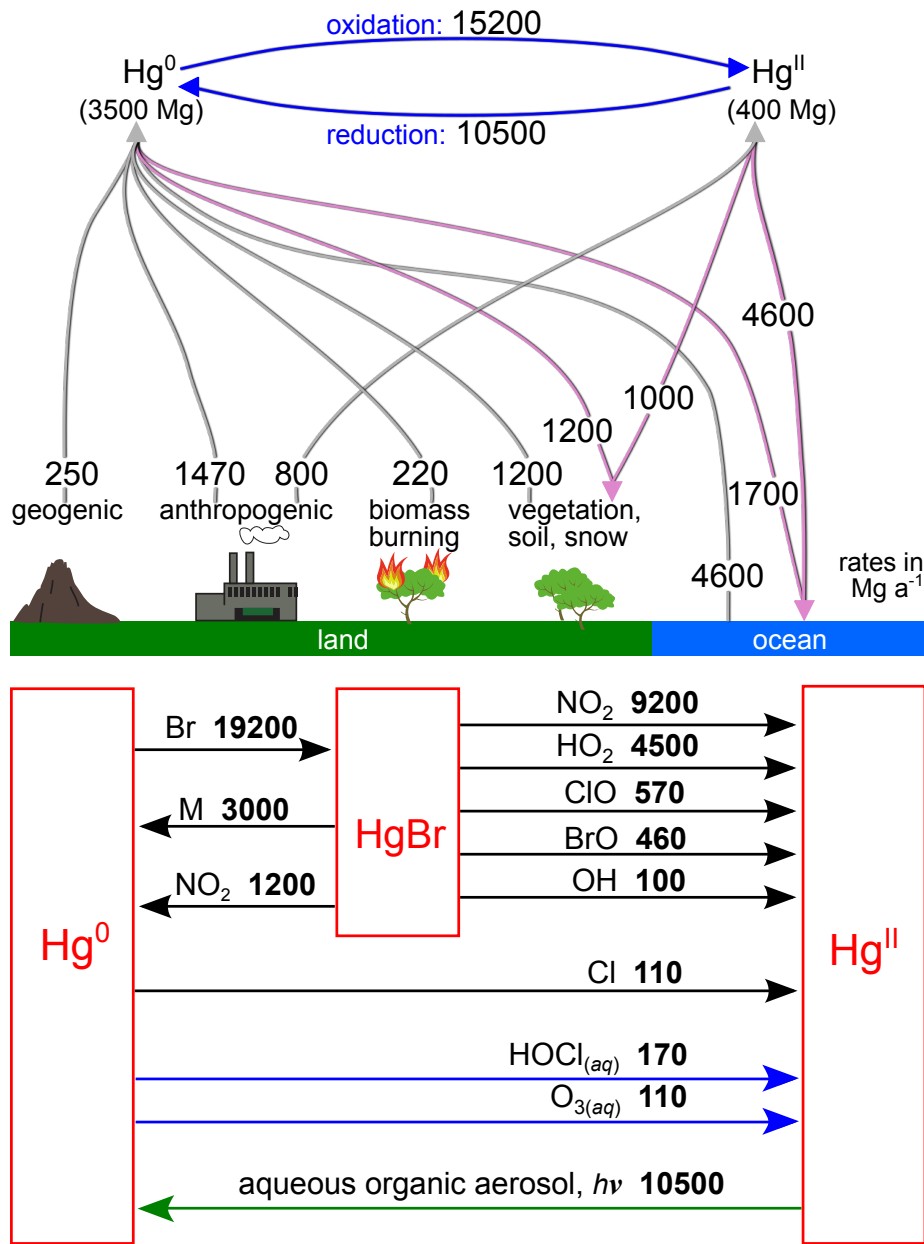

**Figure 2. Global budget of tropospheric mercury in GEOS-Chem. Hg[II] includes gaseous and particulate forms in local equilibrium (Amos et al., 2012). The bottom panel identifies the major chemical reactions from Table 1 cycling Hg[0] and Hg[II]. Reactions with global rates lower than 100 Mg a[-1] are not shown.**



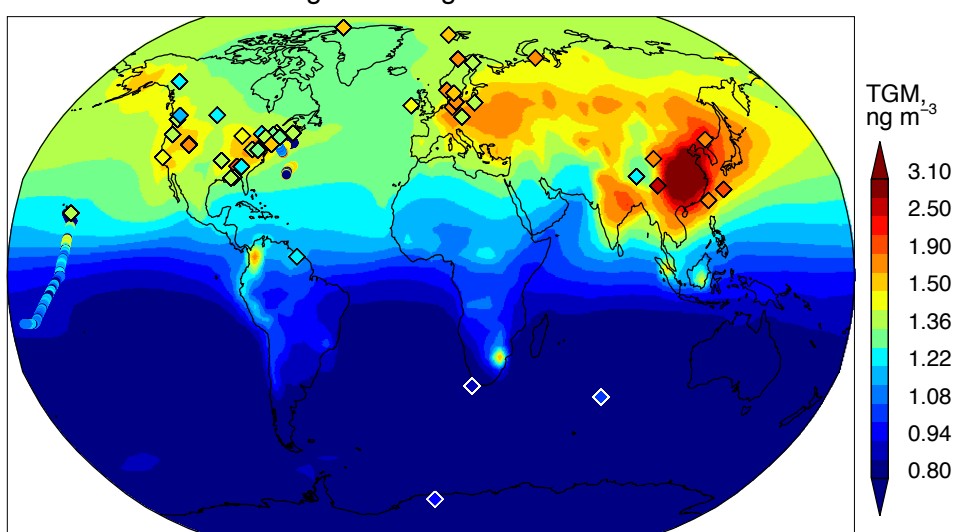

**Figure 3. Global distribution of total gaseous mercury (TGM) concentrations in surface air. Model values (background) are annual means for 2009-2011. Observations (symbols) are for 2007-2013. Data for land sites (diamonds) are annual means for 2007-2013 as previously compiled by Song et al. (2015) and Zhang et al. (2016). Data at three Nordic stations were converted from 20°C to ng m$^{-3}$ STP ($p$ = 1 atm, $T$ = 273 K) (see Slemr et al., 2015). Observations from 2007-2013 ship cruises (circles) are from Soerensen et al. (2013, 2014). Note the change in the linear color scale at 1.50 ng m$^{-3}$.**



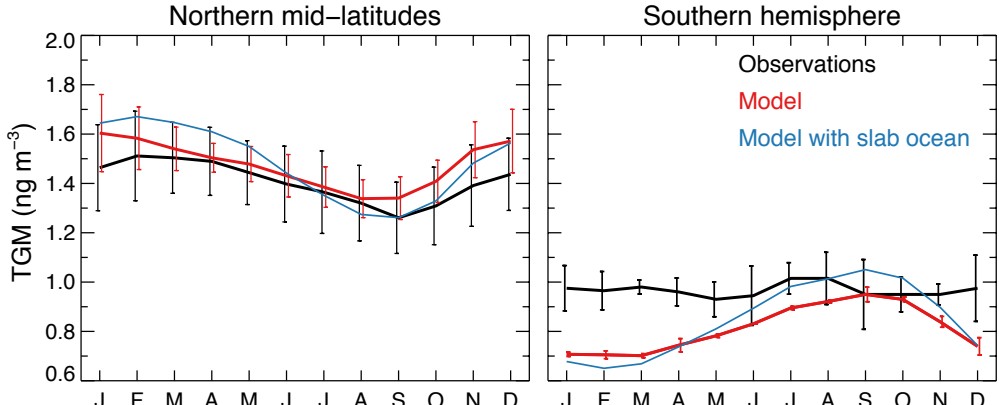

**Figure 4. Mean seasonal variation and spatial standard deviation of total gaseous mercury (TGM) concentrations for northern mid-latitude sites (see Figure 3) and southern hemisphere sites (Amsterdam Island and Cape Point). Observations are compared to results from our standard simulation (based on GEOS-Chem coupled to the MITgcm 3-D ocean model) to a sensitivity study with the 2-D surface-slab ocean model in GEOS-Chem.**





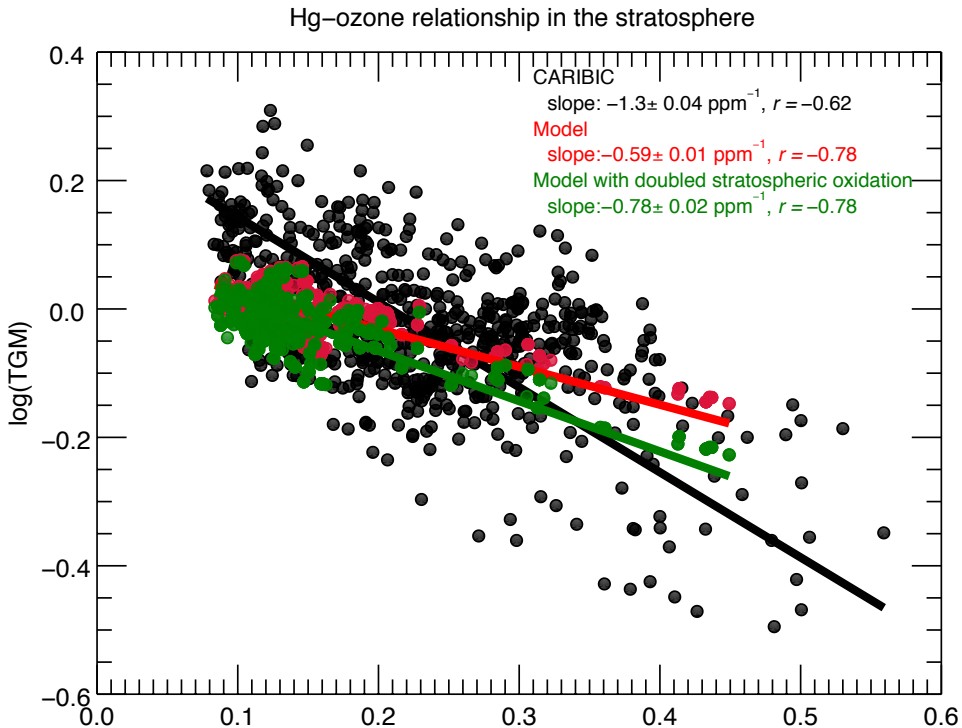

**Figure 5. Relationship between total gaseous mercury (TGM) and ozone concentrations in the lower stratosphere. TGM is shown as the decimal logarithm of the concentration in ng m$^{-3}$ STP. Observations are from CARIBIC commercial aircraft in the extratropical northern hemisphere for April 2014-January 2015 (Slemr et al., 2015). Tropospheric data as diagnosed by [O$_3$]/[CO] < 1.25 mol mol$^{-1}$ are excluded. Model points are data for individual flights sampled along the CARIBIC flight tracks on the GEOS-Chem 4°x5° model grid. Also shown are results from a model sensitivity simulation with a doubled Hg$^0$ oxidation rate in the stratosphere. Reduced-major-axis (RMA) regressions for the log(TGM)-ozone relationship are shown with slopes and correlation coefficients. Errors on the slopes are estimated by the bootstrap method.**





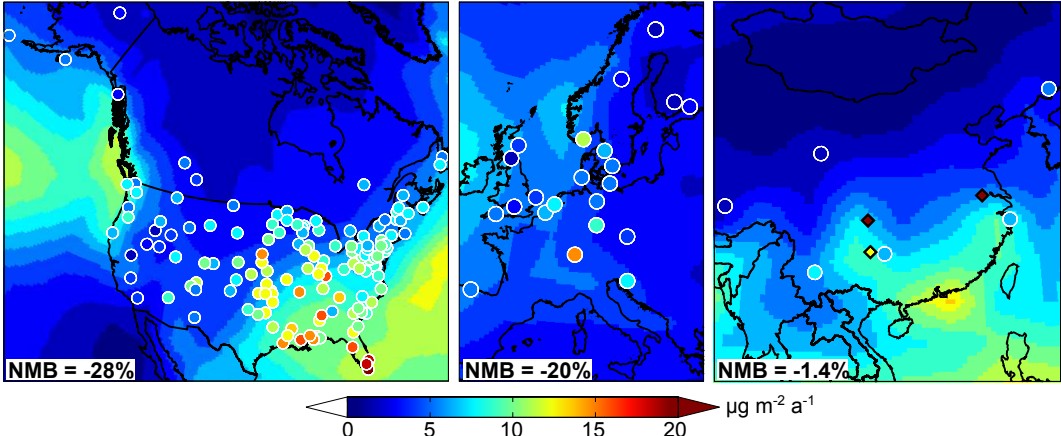

**Figure 6. Annual Hg wet deposition fluxes over North America, Europe, and China. Model values for 2009-2011 (background contours) are compared to 2007-2013 observations from the Mercury Deposition Network (MDN, National Atmospheric Deposition Program, http://nadp.sws.uiuc.edu/mdn/) over North America (58 sites), the European Monitoring and Evaluation Program (EMEP) over Europe (20 sites), and data from Fu et al. (2015, 2016) over China (9 sites). In the China panel, circles represent rural sites and diamonds represent urban sites as identified in Fu et al. (2015, 2016). For the MDN and EMEP networks, which collect weekly or monthly integrated samples, we only include sites with at least 75% of annual data for at least one year between 2007 and 2013; for China we only include sites with at least 9 months of data over the 2007-2013 period. MDN data are formally quality-controlled, while for EMEP data we rely on a subset of sites that have been quality-controlled (Oleg Travnikov, personal communication). Legends give the normalized mean bias (NMB) for all sites, excluding urban sites in China.**





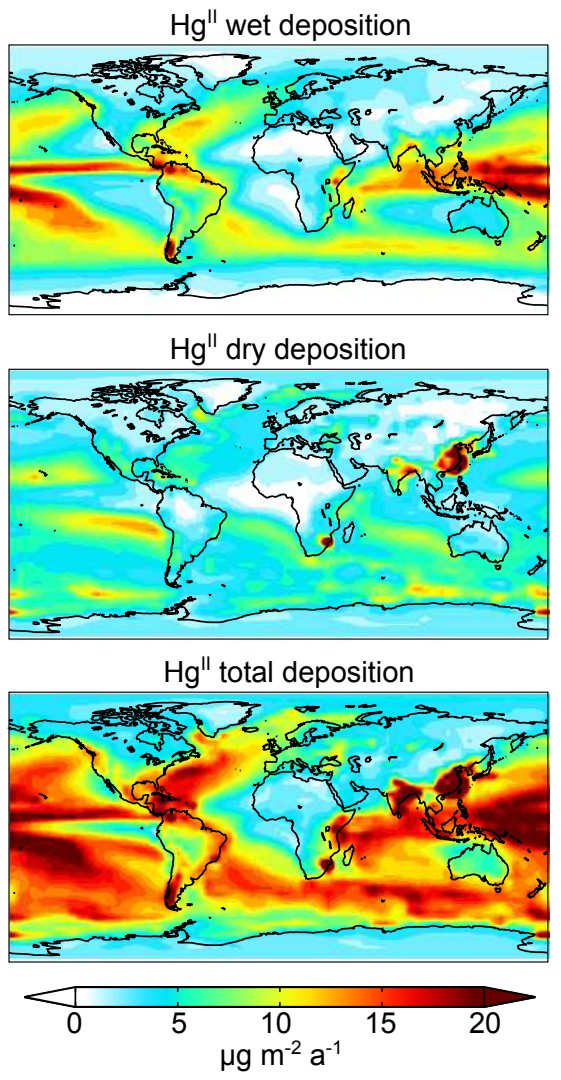

**Figure 7. Annual 2009-2011 Hg<sup>II</sup> deposition fluxes in GEOS-Chem.**