# Peer review of "A new mechanism for atmospheric mercury redox chemistry: Implications for the global mercury budget"

_Atmospheric Chemistry and Physics, 2016_

## Referee Comment (RC1) · I.M. Hedgecock (Referee) · 9 Feb 2017

This article describes recent changes to the chemical mechanism used in the GEOS-Chem model to simulate the atmospheric mercury cycle. The new mechanism includes the recent theoretical results from Dibble et al. concerning the reaction of the unstable HgBr$^*$ intermediate with $HO_2$ and $NO_2$, and also aqueous phase photoreduction of organic Hg complexes 'tuned' to match observed mean total gaseous mercury concentrations and variability. The results are encouraging and suggest that this mechanism may well be a major step forward in the simulation of the atmospheric mercury cycle and provides some interesting new ideas for future measurement campaigns.

[Figure]

I just have a few of comments/suggestions to make.

In the discussion of the unlikelihood of $O_3$ or OH being atmospheric Hg oxidants (Introduction, line 14 onwards), it may be appropriate to point out that homogeneous reactions are being discussed and that the possibility remains that heterogeneous reactions are possible, see Ariya et al (Chem. Rev. 2015, 115, 3760−3802, DOI: 10.1021/cr500667e). In the General Description of the GEOS-Chem model (section 3.1) the gas-particle partitioning of Hg is described as being a local thermodynamic equilibrium dependent on aerosol mass concentration and temperature. This is an approximation as aerosol surface area and composition must also play a role in the partitioning, for instance one would expect soot particles to adsorb Hg rather effectively and be less sensitive to changes in temperature. This should be pointed out. In section 3.2 the sources of radical, oxidant and organic aerosol concentrations are mentioned, but there is no mention of how these modelled concentrations compared to available measurement data, ground based or otherwise. The Schmidt et al. paper describes the comparison of BrO with satellite data but some mention of how $NO_2$ and organic aerosol compare or references to where such comparisons may be found would be useful. I see the difficulty in comparing modelled $HO_2$. However some comparison is important as the oxidation of Hg in the model is dominated by the reaction of HgBr$^*$ by $NO_2$ and $HO_2$.

In Section 4.1 the authors state that the vertical structure of the Hg$^0$ concentration is *well* established, when in reality there have actually been only a few measurements made, given that CARIBIC measures TGM, perhaps this an overstatement.

In Section 4.3 the seasonality is discussed, and states (line 27) that oxidation is faster in the summer than the winter. However the lower temperatures in the winter increase the lifetime against thermal decomposition of HgBr$^*$, so perhaps a little more detail in the explanation would be useful. Also here one assumes that oceanic evasion is highest in the winter due to less clement weather, but it is not explicitly stated. On the same subject, in the Conclusions (line 18) oxidation and evasion are described as

having similar seasonal phases, but anticorrelated?

In Section 5, the authors discuss the high Hg deposition over the Gulf of Mexico, the figures show that modelled deposition is higher here but it is still less than the observed values. Is this due to the model resolution and reproducing sub-grid scale convection?

The article is well written and merits publication in ACP.

---

## Referee Comment (RC2) · Anonymous Referee #2 · 9 Feb 2017

This is a great paper that represents great progress in the field of atmospheric chemistry. It definitely deserves publication. I have a few specific comments:

1. On page 2 lines 14-15, you talk about how oxidation by O3 and OH is unlikely. It is important, I think, that you bring out and discuss (or debunk, if they merit it) other oxidation/reduction mechanisms that have been proposed, including heterogeneous mechanisms. No one disputes that halogen-involving reactions are important. But there are some strong voices that claim other mechanisms may also be important, and I think it would be useful to give them consideration.

2. On page 2 lines 30-32, you suggest that in-plume reduction isn't important, and give the Deeds and Landis papers as references. Both papers show evidence for in-plume

reduction, and the Landis paper talks about how the process is likely widespread. Maybe I am misinterpreting your text or theirs, but I'm not sure the statement you make here is supportable by the evidence you have given.

3. I am curious about why in Figure 1 the model shows quite high annual mean HgˆII at the surface near the south pole and overall shows much higher HgˆII in the southern troposphere than in the northern. I think this is a surprising result, and I don't think it was discussed in the manuscript. I would be glad to see some explicit discussion of the reason for the north-south difference in HgˆII concentrations. Unfortunately, we have very few (if any) reliable HgˆII measurements with which to validate this result.

4. Page 7 lines 8-10: again, some discussion of how inclusion of heterogeneous oxidation chemistry might play a role could be useful here.

5. Page 10 lines 1-5: It is not at all clear to me that these modifications to GEOS-Chem "solve" the problem of underprediction of wet deposition in the Gulf of Mexico. Based on Figure 6, the model is still strongly underestimating Gulf of Mexico wet deposition. I don't think the model even gets the basic spatial trends right, based on my visual inspection of Figure 6. It may be fair to say that it is doing better than before, but to say it has it all nailed down seems to be an overstatement.

6. Page 11 lines 30-31: I don't agree that the "wet deposition fluxes in the model are consistent with observations." This assertion needs to be toned down.

---

## Referee Comment (RC3) · Anonymous Referee #3 · 17 Feb 2017

In the presented paper the authors investigate the implications of two stage oxidation mechanisms for the global mercury cycle. This work is highly innovative and will have a strong impact on atmospheric mercury modelling and our understanding on mercury red-ox reactions. Thus, I strongly support publication of this manuscript after some minor revisions:

P1 L18: "Hg2+ controls deposition to ecosystems" I think that this is not 100% correct as it neglects the importance of Hg0 dry deposition. (e.g. Zhang, L., Blanchard, P., Gay, D.A., Presbo, E.M., Risch, M.R., Johnson, D., Narayan, J., Zsolway, R., Holsen, T.M., Miller, E.K., Castro, M.S., Graydon, J.A., St. Louis, V.L., Dalziel, J., (2012a). Estimation of speciated and total mercury dry deposition at monitoring locations in

eastern and central North America. ACP 12, 4327-4340.)

P1 L 33: "lowermost stratosphere show a strong TGM-ozone relationship" I want to add to that while this is true for many observations, that during some CARIBIC flights no such correlation was found. As Franz Slemr is a co-author I think he knows best how to phrase this correctly.

P2 L32: For completeness, I would like to add another recent aircraft observation on the issue with similar findings: (Weigelt, A., 2016. Mercury emissions of a coal-fired power plant in Germany. Atmospheric Chemistry and Physics 16(21):13653-13668. doi: 10.5194/acp-16-13653-2016)

P3 L35: I would appreciate if you would give a complete list of molecules considered as reaction partner Y. Such that the paragraph is consistent with Table 2 where you list (Y = HO2, OH, Cl, BrO, ClO).

P4 Section 3.1: You compare the recent model results from findings from Holmes et al. 2010. Here, I am missing a more detailed comparison of the Br fields used for both models as this is one major driver for the model results.

P5 L 23ff: Atmospheric models are still having problems reproducing atmospheric OA concentrations mainly due to an underestimation of SOA formation. When using OA concentrations for Hg reduction processes you need to clarify the quality of the OA fields used for this purpose. Does the correction parameter alpha you use compensate for too low OA concentrations? Or does the model use increases POA emissions to compensate SOA formation?

P6 L 11-15: I understand that in this paper you focus on the troposphere and thus include only a limited discussion of the stratosphere. But in my opinion (based on CARIBIC observations and a multi model study on mercury vertical distribution) the way you phrase this is not 100% correct. Hg2+ concentration strongly increases in the stratophere. But most models as well as the observations do not indicate a domination

(>50%) of Hg2+.

You might be aware of this paper also currently under review in ACPD? (Bieser, J., Slemr, F., Brenningkmeijer, K., Brooks, S., Dastoor A., De Simone, F., Ebinghaus R., Gencarelli, C.N., Geyer, B., Gratz, L., Hedgecock, I.M., Jaffe, D., Kelley, P., Lin C.-C., Matthias, V., Selin, N., Shah, V., Song, S., Travnikov, O., Weigelt, A., Winston, L., Zahn, A., Pirrone, N., Multi-model study of mercury dispersion in the atmosphere: Vertical distribution of mercury species. Atmos. Chem. Phys. Disc. doi:10.5194/acp-2016-1074 2016.)

P7 L2: I am confused by this estimated lifetime of 1.2 to 2.8 months. It is contradictory to the generally expected life time close to 6 months (which is the value you mention later on and with which I agree). I think you need to put the number from Shah et al. 2016 more into context here. e.g. During the NOMADSS campaign they found a few episodes with very high bromine concentrations. Do they dominate this estimation?

P7 L25: (Similar to last) Please clarify: 1) That the "NOMADSS simulation" is also a GEOS-Chem simulation and indicate which version, chemsitry, and br fields were used. 2) That the NOMADSS campaign was limited to certain latitudes of the US (do you compare only life times for this region?)

P7 L28-29: This is a very important part of your results but it seems to be circular reasoning and I would appreciate it if you could clarify this: As far as I understand you have high bromine concentrations leading to high first stage oxidation rates and a high actinic flux leading to high secondary oxidation and thus need a higher reduction rate to archive conformity between model and observations. However, could you not also conclude that you are overestimating oxidation rates or bromine concentrations rather than underestimating reduction rates? So the question seems: Is reduction an actual sink for Hg2+ or could the red-ox reaction also be understood as steady state condition, and thus as a tuning parameter in the free troposphere? Or are there specific temporal or spatial signals in the observations which are only reproduced by your mechanism?

Moreover, it would be interesting if you could give more qualitative statements on the distribution of the reduction in the free troposphere. As your models needs particles, organic aerosols, and liquid water I assume that the reduction is not evenly distributed in the free troposphere. I would actually expect a higher reduction potential inside the PBL due to high concentrations of particles and liquid water. Also how do you treat ice particles?

P8 L 19: Would this also be true for the inter-hemispheric gradient observed in high altitudes (CARIBIC data)?

P10 L10-16: I just wanted to underline how important this finding is because of the impact on long range transport of mercury from east Asia.

P10 L24-17: Can you give the net flux between atmosphere and ocean?

P11 L 23-29: Is the model able to reproduce episodes where no TGM/ozone relationship was observed by CARABIC?

Minor remarks concerning language:

P4 L8: for for (double)

P1 L18: In case my sense for punctuation is correct there is a , missing after "Here".

P1 L33: southern hemispheric marine sites, not southern hemisphere marine sites.

---

## Referee Comment (RC4) · Anonymous Referee #4 · 20 Feb 2017

This paper describes advancements in the modeling of global Hg cycling through updated redox chemistry as well as coupling to a more sophisticated ocean model. The approach is thorough and defensible and the paper well-written. I recommend publication after minor revisions.

P5, l. 23-25: Can you add a summary of how accurate these OA concentrations are expected to be? If this isn't known, how does that impact your conclusions?

P8, l. 1-20: This discussion of the mean and SD of TGM concentrations underplays the spatial deviations shown in Fig. 3. For example, the modeled and measured N-S gradients in Europe appear to be in opposite directions.

P8, l. 28-30: I suggest you cite Soerenson et al., EST 2013, or other observation-based papers to show that this mechanism is consistent with observations.

P9, l. 5-7: It sounds like you are suggesting that a similar analysis for the southern ocean is needed, but both the sea ice and the relatively large riverine flux relative to the Hg pool in the Arctic Ocean are not likely to have any relevance to these southern non-polar sites.

P10, l. 1-5: The figure indicates a -28% mean bias in North America, but in the Gulf of Mexico it appears to still be closer to -50%. Can you quantify the improvement in the wet deposition relative to Holmes et al. (or other models)? Consider adding a figure showing scatterplots of model vs. observations for both TGM and wet deposition.

---

## Author Comment (AC1) · 20 Apr 2017

Thank you for your thoughtful comments. Including your suggested revisions has improved the quality of the manuscript. Our responses are indicated below in blue text.

This article describes recent changes to the chemical mechanism used in the GEOSChem model to simulate the atmospheric mercury cycle. The new mechanism includes the recent theoretical results from Dibble et al. concerning the reaction of the unstable HgBr∗ intermediate with HO2 and NO2, and also aqueous phase photoreduction of organic Hg complexes 'tuned' to match observed mean total gaseous mercury concentrations and variability. The results are encouraging and suggest that this mechanism may well be a major step forward in the simulation of the atmospheric mercury cycle and provides some interesting new ideas for future measurement campaigns.

I just have a few of comments/suggestions to make. In the discussion of the unlikelihood of O3 or OH being atmospheric Hg oxidants (Introduction, line 14 onwards), it may be appropriate to point out that homogeneous reactions are being discussed and that the possibility remains that heterogeneous reactions are possible, see Ariya et al (Chem. Rev. 2015, 115, 3760−3802, DOI: 10.1021/cr500667e).

We add a sentence (Page 2, lines 19-20):

"OH and ozone could still potentially be important oxidants on aerosols (Ariya et al., 2015)."

In the General Description of the GEOS-Chem model (section 3.1) the gas-particle partitioning of Hg is described as being a local thermodynamic equilibrium dependent on aerosol mass concentration and temperature. This is an approximation as aerosol surface area and composition must also play a role in the partitioning, for instance one would expect soot particles to adsorb Hg rather effectively and be less sensitive to changes in temperature. This should be pointed out.

We clarify the description of the gas-particle partitioning and add a sentence (page 4, lines 26-30):

"Gas-particle partitioning of $Hg^{II}$ is parameterized following Amos et al. (2012) as a thermodynamic equilibrium function of local temperature and mass concentration of fine particulate matter ($PM_{2.5}$). This parameterization is based on observed relationships between the gas/particle $Hg^{II}$ concentration ratio and $PM_{2.5}$, and does not resolve effects from particle composition or surface area."

In section 3.2 the sources of radical, oxidant and organic aerosol concentrations are mentioned, but there is no mention of how these modelled concentrations compared to available measurement data, ground based or otherwise. The Schmidt et al. paper describes the comparison of BrO with satellite data but some mention of how NO2 and organic aerosol compare or references to where such comparisons may be found would be useful. I see the difficulty in comparing modelled HO2. However some comparison is important as the oxidation of Hg in the model is dominated by the reaction of HgBr∗ by NO2 and HO2.

We add the following text to this section (page 5, lines 32-36):

"GEOS-Chem concentrations of $NO_2$ and $H_2O_2$ (proxy for $HO_2$) have been evaluated successfully in a number of aircraft campaigns (Martin et al., 2006; Hudman et al., 2007; Singh et al., 2007; Lin and McElroy, 2010; Mao et al., 2010; Travis et al., 2016). GEOS-Chem $NO_2$ columns have also been evaluated against satellite observations over China (Lin et al., 2012), North America (e.g., Lamsal et al., 2014), and Africa (Marais et al., 2012).

We also add a sentence in the same paragraph regarding organic aerosol (page 6, line 3-5): "An evaluation of modeled OA against aircraft observations globally is presented in Heald et al. (2011), which showed no systematic bias in remote environments but an underestimate of median concentrations in polluted regions."

In Section 4.1 the authors state that the vertical structure of the Hg0 concentration is well established, when in reality there have actually been only a few measurements made, given that CARIBIC measures TGM, perhaps this an overstatement.
We revise this paragraph and add clarifying text as follows (page 6, lines 30 to 36) to address this and other reviewers' comments:
"Modeled $Hg^0$ decreases rapidly in the stratosphere, while $Hg^{II}$ increases with altitude and dominates total Hg in the stratosphere. This vertical structure is driven by chemistry (Selin et al., 2007; Holmes et al., 2010) and these general vertical trends in the two species are consistent with limited available observations (Murphy et al., 2006; Talbot et al., 2007; Lyman and Jaffe, 2012). The exact partitioning of total Hg in the stratosphere between $Hg^0$ and $Hg^{II}$ is uncertain and GEOS-Chem predicts a higher $Hg^{II}$ fraction relative to other models (Bieser et al., 2016). Photodissociation of gas-phase $Hg^{II}$ halides may be possible at ultraviolet wavelengths (Maya, 1977) but whether this is important in the mid-to-upper stratosphere requires further investigation."

In Section 4.3 the seasonality is discussed, and states (line 27) that oxidation is faster in the summer than the winter. However the lower temperatures in the winter increase the lifetime against thermal decomposition of HgBr∗ , so perhaps a little more detail in the explanation would be useful. Also here one assumes that oceanic evasion is highest in the winter due to less clement weather, but it is not explicitly stated.
We clarify text in the first paragraph of Section 4.3:
"The February maximum and September minimum are driven in the model in part by $Hg^0$ oxidation and in part by ocean evasion. Seasonality in radical oxidant concentrations outweighs temperature effects, leading to fastest oxidation in summer and slowest in winter. Modeled oceanic $Hg^0$ evasion from the North Atlantic Ocean peaks in winter and early spring, because higher windspeeds increase mixing and entrainment of reducible $Hg^{II}$ from subsurface waters and enhance the rate of air-sea exchange, and Hg removal via particle settling during this time is low (Sorensen et al., 2010, 2013)."

On the same subject, in the Conclusions (line 18) oxidation and evasion are described as paper having similar seasonal phases, but anticorrelated?
We reword this sentence in line 20 of page 12 to clarify:
"…with similar seasonal effects on TGM concentrations."

In Section 5, the authors discuss the high Hg deposition over the Gulf of Mexico, the figures show that modelled deposition is higher here but it is still less than the observed values. Is this due to the model resolution and reproducing sub-grid scale convection?
We rewrite this paragraph (now page page 10, lines 34-36 to page 11, lines 1-2) to address your and other reviewers' comments:
"Previous GEOS-Chem simulations with Br-initiated oxidation of $Hg^0$ failed to capture this maximum because $Hg^{II}$ production favored higher latitudes (Holmes et al., 2010; Amos et al.,

2012). The inclusion of $NO_2$ and $HO_2$ as second-stage oxidants in our simulation shifts $Hg^{II}$ production to lower latitudes and matches the general location of the Gulf of Mexico maximum although the magnitude is still underestimated. Increasing horizontal resolution in the model could improve the definition of the maximum (Zhang et al., 2012)."

The article is well written and merits publication in ACP.

---

## Author Comment (AC2) · 20 Apr 2017

Thank you for your thoughtful comments. Including your suggested revisions has improved the quality of the manuscript. Our responses are indicated below in blue text.

This is a great paper that represents great progress in the field of atmospheric chemistry. It definitely deserves publication. I have a few specific comments:
1. On page 2 lines 14-15, you talk about how oxidation by O3 and OH is unlikely. It is important, I think, that you bring out and discuss (or debunk, if they merit it) other oxidation/reduction mechanisms that have been proposed, including heterogeneous mechanisms. No one disputes that halogen-involving reactions are important. But there are some strong voices that claim other mechanisms may also be important, and I think it would be useful to give them consideration.
We add a sentence (Page 2, lines 19-20):
"OH and ozone could still potentially be important oxidants on aerosols (Ariya et al., 2015)."

We chose to leave the more detailed discussion of other proposed mechanisms in Section 2 and Tables 1 and 2 instead of in the introduction.

2. On page 2 lines 30-32, you suggest that in-plume reduction isn't important, and give the Deeds and Landis papers as references. Both papers show evidence for in-plume reduction, and the Landis paper talks about how the process is likely widespread. Maybe I am misinterpreting your text or theirs, but I'm not sure the statement you make here is supportable by the evidence you have given.

We revised this sentence for clarity and added text (page 2 lines 33 to 37, page 3 lines 1) as follows:

"Fast in-plume reduction of $Hg^{II}$ emitted by coal-fired power plants was first reported by Edgerton et al. (2006) and Lohman et al. (2006) but more recent field observations suggest that on average only 5% (range 0-55%) of emitted $Hg^{II}$ is reduced in the plume (Deeds et al., 2013; Landis et al., 2014). Recent aircraft observations (Weigelt et al., 2016) and emission inventories (Zhang et al., 2016) suggest that previous reports of in-plume reduction of $Hg^{II}$ may reflect in part an overestimate of $Hg^{II}$ emissions. "

3. I am curious about why in Figure 1 the model shows quite high annual mean HgˆII at the surface near the south pole and overall shows much higher HgˆII in the southern troposphere than in the northern. I think this is a surprising result, and I don't think it was discussed in the manuscript. I would be glad to see some explicit discussion of the reason for the north-south difference in HgˆII concentrations. Unfortunately, we have very few (if any) reliable HgˆII measurements with which to validate this result.

We add the following discussion to Section 4.1 (page 8 lines 18-20).
"$Hg^{II}$ reduction in the model is faster in the northern hemisphere than in the southern hemisphere because of higher OA concentrations. Hence $Hg^{II}$ concentrations are higher in the southern than in the northern hemisphere (Figure 1)."

4. Page 7 lines 8-10: again, some discussion of how inclusion of heterogeneous oxidation chemistry might play a role could be useful here.

We add the following sentences (now page 7, lines 34-37):
"We did not consider heterogeneous $Hg^0$ oxidation on particle surfaces (e.g., Vidic et al., 1998; Flora et al., 1998; Lee et al., 2004) due to inadequate information to formulate atmospheric rates. Further study of heterogeneous $Hg^0$ oxidation is needed (Ariya et al., 2015)."

5. Page 10 lines 1-5: It is not at all clear to me that these modifications to GEOS-Chem "solve" the problem of underprediction of wet deposition in the Gulf of Mexico. Based on Figure 6, the model is still strongly underestimating Gulf of Mexico wet deposition. I don't think the model even gets the basic spatial trends right, based on my visual inspection of Figure 6. It may be fair to say that it is doing better than before, but to say it has it all nailed down seems to be an overstatement.

We rewrite this paragraph (now page page 10, lines 34-36 to page 11, lines 1-2) to address your and other reviewers' comments:
"Previous GEOS-Chem simulations with Br-initiated oxidation of $Hg^0$ failed to capture this maximum because $Hg^{II}$ production favored higher latitudes (Holmes et al., 2010; Amos et al., 2012). The inclusion of $NO_2$ and $HO_2$ as second-stage oxidants in our simulation shifts $Hg^{II}$ production to lower latitudes and matches the general location of the Gulf of Mexico maximum although the magnitude is still underestimated. Increasing horizontal resolution in the model could improve the definition of the maximum (Zhang et al., 2012)."

6. Page 11 lines 30-31: I don't agree that the "wet deposition fluxes in the model are consistent with observations." This assertion needs to be toned down.

We revised this and the following sentences as follows (page 12, lines 32-36):
"Hg wet deposition fluxes in the model are consistent with spatial patterns observed in North America and Europe. Inclusion of $NO_2$ and $HO_2$ as second-stage HgBr oxidants in the model shifts $Hg^{II}$ production to lower latitudes compared to previous versions of GEOS-Chem and enables the model to capture the location of the observed maximum in wet deposition along the Gulf Coast of the US. However, the magnitude of this Gulf Coast maximum is still underestimated."

We also modified the Abstract and Section 5 text, which are consistent with the above.

---

## Author Comment (AC3) · 20 Apr 2017

Thank you for your thoughtful comments. Including your suggested revisions has improved the quality of the manuscript. Our responses are indicated below in blue text.

In the presented paper the authors investigate the implications of two stage oxidation mechanisms for the global mercury cycle. This work is highly innovative and will have a strong impact on atmospheric mercury modelling and our understanding on mercury red-ox reactions. Thus, I strongly support publication of this manuscript after some minor revisions:

P1 L18: "Hg2+ controls deposition to ecosystems" I think that this is not 100% correct as it neglects the importance of Hg0 dry deposition. (e.g. Zhang, L., Blanchard, P., Gay, D.A., Presbo, E.M., Risch, M.R., Johnson, D., Narayan, J., Zsolway, R., Holsen, T.M., Miller, E.K., Castro, M.S., Graydon, J.A., St. Louis, V.L., Dalziel, J., (2012a). Estimation of speciated and total mercury dry deposition at monitoring locations in eastern and central North America. ACP 12, 4327-4340.)
We modify page 1 lines 17 to 18: " Oxidation to water-soluble $Hg^{II}$ plays a major role in Hg deposition to ecosystems."

P1 L 33: "lowermost stratosphere show a strong TGM-ozone relationship" I want to add to that while this is true for many observations, that during some CARIBIC flights no such correlation was found. As Franz Slemr is a co-author I think he knows best how to phrase this correctly.
In this paper we do not compare on a flight-by-flight basis but to the overall relationship across all flights.

P2 L32: For completeness, I would like to add another recent aircraft observation on the issue with similar findings: (Weigelt, A., 2016. Mercury emissions of a coal-fired power plant in Germany. Atmospheric Chemistry and Physics 16(21):13653-13668. doi: 10.5194/acp-16-13653-2016)
We edit this section, (page 2 lines 33 to 37, page 3 lines 1) as follows:
"Fast in-plume reduction of $Hg^{II}$ emitted by coal-fired power plants was first reported by Edgerton et al. (2006) and Lohman et al. (2006) but more recent field observations suggest that on average only 5% (range 0-55%) of emitted $Hg^{II}$ is reduced in the plume (Deeds et al., 2013; Landis et al., 2014). Recent aircraft observations (Weigelt et al., 2016) and emission inventories (Zhang et al., 2016) suggest that previous reports of in-plume reduction of $Hg^{II}$ may reflect in part an overestimate of $Hg^{II}$ emissions. "

P3 L35: I would appreciate if you would give a complete list of molecules considered as reaction partner Y. Such that the paragraph is consistent with Table 2 where you list (Y = HO2, OH, Cl, BrO, ClO).
We clarify the sentence, now page 4, lines 2 to 4:
"Dibble et al. (2012) found that a broad range of radical oxidants could oxidize HgBr and HgCl including $Y \equiv NO_2$ and $HO_2$, the most abundant atmospheric radicals, as well as $Y \equiv BrO$, ClO, and Cl."

P4 Section 3.1: You compare the recent model results from findings from Holmes et al. 2010. Here, I am missing a more detailed comparison of the Br fields used for both models as this is one major driver for the model results.

We add text to point the reader to section 3.2, page 5, lines 1-2: "In this work we update the Br concentration fields to a more recent version of GEOS-Chem (Schmidt et al., 2016), as discussed further in Section 3.2."

P5 L 23ff: Atmospheric models are still having problems reproducing atmospheric OA concentrations mainly due to an underestimation of SOA formation. When using OA concentrations for Hg reduction processes you need to clarify the quality of the OA fields used for this purpose. Does the correction parameter alpha you use compensate for too low OA concentrations? Or does the model use increases POA emissions to compensate SOA formation?
We add a sentence in this section so it now reads as follows (page 6, lines 1-5):
"Monthly mean organic aerosol (OA) concentrations are archived from a separate v9-02 GEOS-Chem simulation including primary emissions from combustion and secondary production from biogenic and anthropogenic hydrocarbons (Pye et al., 2010). An evaluation of modeled OA against aircraft observations globally is presented in Heald et al. (2011), which showed no systematic bias in remote environments but an underestimate of median concentrations in polluted regions."

P6 L 11-15: I understand that in this paper you focus on the troposphere and thus include only a limited discussion of the stratosphere. But in my opinion (based on CARIBIC observations and a multi model study on mercury vertical distribution) the way you phrase this is not 100% correct. Hg2+ concentration strongly increases in the stratophere. But most models as well as the observations do not indicate a domination (>50%) of Hg2+.
You might be aware of this paper also currently under review in ACPD? (Bieser, J., Slemr, F., Brenningkmeijer, K., Brooks, S., Dastoor A., De Simone, F., Ebinghaus R., Gencarelli, C.N., Geyer, B., Gratz, L., Hedgecock, I.M., Jaffe, D., Kelley, P., Lin C.- C., Matthias, V., Selin, N., Shah, V., Song, S., Travnikov, O., Weigelt, A., Winston, L., Zahn, A., Pirrone, N., Multi-model study of mercury dispersion in the atmosphere: Vertical distribution of mercury species. Atmos. Chem. Phys. Disc. doi:10.5194/acp- 2016-1074 2016.)

We revise this paragraph and add clarifying text as follows (page 6, lines 30 to 36) to address this and other reviewers' comments:
"Modeled $Hg^0$ decreases rapidly in the stratosphere, while $Hg^{II}$ increases with altitude and dominates total Hg in the stratosphere. This vertical structure is driven by chemistry (Selin et al., 2007; Holmes et al., 2010) and these general vertical trends in the two species are consistent with limited available observations (Murphy et al., 2006; Talbot et al., 2007; Lyman and Jaffe, 2012). The exact partitioning of total Hg in the stratosphere between $Hg^0$ and $Hg^{II}$ is uncertain and GEOS-Chem predicts a higher $Hg^{II}$ fraction relative to other models (Bieser et al., 2016). Photodissociation of gas-phase $Hg^{II}$ halides may be possible at ultraviolet wavelengths (Maya, 1977) but whether this is important in the mid-to-upper stratosphere requires further investigation."

P7 L2: I am confused by this estimated lifetime of 1.2 to 2.8 months. It is contradictory to the generally expected life time close to 6 months (which is the value you mention later on and with which I agree). I think you need to put the number from Shah et al. 2016 more into context here. e.g. During the NOMADSS campaign they found a few episodes with very high bromine concentrations. Do they dominate this estimation?

P7 L25: (Similar to last) Please clarify: 1) That the "NOMADSS simulation" is also a GEOS-Chem simulation and indicate which version, chemsitry, and br fields were used. 2) That the NOMADSS campaign was limited to certain latitudes of the US (do you compare only life times for this region?)

Here we address the previous two comments.

We add a sentence to clarify the different lifetime estimates (page 7, lines 18 to 20):

"We calculate several atmospheric lifetimes to understand the processes driving Hg deposition: the chemical lifetime of $Hg^0$ against oxidation, the chemical lifetime of $Hg^{II}$ against reduction, the lifetime of $Hg^{II}$ against deposition, and the lifetime of total gaseous mercury (TGM ≡ $Hg^0 + Hg^{II}(g)$) against deposition."

We also clarify the discussion of Shah et al. in that paragraph (page 7, lines 25 to 28):

"Our results are consistent with Shah et al. (2016), who estimated a global annual tropospheric $Hg^0$ lifetime against oxidation to $Hg^{II}$ by Br atoms of 1.2 to 2.8 months, based on their simulation of measurements of $Hg^{II}$ over the Southeast US in summer during the NOMADSS campaign (Gratz et al., 2015)."

P7 L28-29: This is a very important part of your results but it seems to be circular reasoning and I would appreciate it if you could clarify this: As far as I understand you have high bromine concentrations leading to high first stage oxidation rates and a high actinic flux leading to high secondary oxidation and thus need a higher reduction rate to archive conformity between model and observations. However, could you not also conclude that you are overestimating oxidation rates or bromine concentrations rather than underestimating reduction rates? So the question seems: Is reduction an actual sink for Hg2+ or could the red-ox reaction also be understood as steady state condition, and thus as a tuning parameter in the free troposphere? Or are there specific temporal or spatial signals in the observations which are only reproduced by your mechanism?

Moreover, it would be interesting if you could give more qualitative statements on the distribution of the reduction in the free troposphere. As your models needs particles, organic aerosols, and liquid water I assume that the reduction is not evenly distributed in the free troposphere. I would actually expect a higher reduction potential inside the PBL due to high concentrations of particles and liquid water. Also how do you treat ice particles?

Here we respond to the above two comments:

Second-stage oxidation rates are high due to the high concentrations of second-stage oxidants. The photoreduction rate of $Hg^{II}$ is a function of actinic flux, but is a parameterization rather than an actual mechanism; the specific $Hg^{II}$ reduction reactions and their rates are unknown. Reduction does not occur on ice particles.

We add a sentence about the evaluation of modeled Br fields in Section 3.2 (page 5, lines 30-32): "Schmidt et al. (2016) evaluated their simulated tropospheric BrO concentrations (global daytime mean of 0.96 ppt) with observations from satellite and aircraft and found no systematic bias."

In the section described in your above comment, we add several sentences and split the text into two paragraphs for clarity, and add text describing the latitudinal distribution of $Hg^{II}$ reduction (page 8, lines 15-28; including sections of the paragraph which are unchanged for convenience, with additions underlined):

"Shah et al. (2016) similarly found that faster $Hg^0$ oxidation as needed to match NOMADSS observations required faster $Hg^{II}$ reduction, with a tropospheric $Hg^{II}$ lifetime against reduction of 19 days. The lifetime of tropospheric $Hg^{II}$ against deposition is relatively long, 26 days (see Figure 2), because most $Hg^{II}$ production occurs in the free troposphere where precipitation is infrequent (Figure 1). $Hg^{II}$ reduction in the model is faster in the northern hemisphere than in the southern hemisphere because of higher OA concentrations. Hence $Hg^{II}$ concentrations are higher in the southern than in the northern hemisphere (Figure 1).

We find here that the $Hg^{II}$ lifetime against reduction is shorter than against deposition, emphasizing the importance of reduction in controlling the atmospheric Hg budget. By contrast, Holmes et al. (2010) found an adjusted $Hg^{II}$ tropospheric lifetime of 50 days against reduction and 36 days against deposition, which led them to conclude that no reduction was needed if $Hg^0$ oxidation kinetics were decreased within their uncertainty. This is no longer possible in our simulation because of the much faster $Hg^0$ oxidation. We conclude that $Hg^{II}$ reduction must take place in the atmosphere. With $Hg^{II}$ reduction, the overall lifetime of tropospheric TGM against deposition in our simulation is 5.2 months, similar to the estimate of 6.1 months in Holmes et al. (2010). We discuss the consistency of this estimate with observations in the next section."

P8 L 19: Would this also be true for the inter-hemispheric gradient observed in high altitudes (CARIBIC data)?
This is a great point but beyond the scope of the current study. We modify page 9 lines 3 to 6 to clarify it is for surface data:
"However, we find the interhemispheric gradient in surface concentrations is not a sensitive diagnostic of lifetime because surface atmospheric Hg in the southern hemisphere is controlled more by atmosphere-ocean exchange than by transport from the northern hemisphere."

P10 L10-16: I just wanted to underline how important this finding is because of the impact on long range transport of mercury from east Asia.
We add a sentence (page 11, lines 13 to 14): "Decreased wet deposition in this region has implications for the long-range transport of atmospheric Hg from East Asia (Weiss-Penzias et al., 2007)."

P10 L24-17: Can you give the net flux between atmosphere and ocean?
We add to the text (page 11, lines 24-26):
"The global ocean is a net sink for total atmospheric Hg of 1700 Mg $a^{-1}$ (Figure 2), with approximately half taken up by tropical oceans."

P11 L 23-29: Is the model able to reproduce episodes where no TGM/ozone relation- ship was observed by CARABIC?
In this paper we do not compare on a flight-by-flight basis but to the overall relationship across all flights.

Minor remarks concerning language: P4 L8: for for (double) P1 L18: In case my sense for

punctuation is correct there is a , missing after "Here". P1 L33: southern hemispheric marine sites, not southern hemisphere marine sites.

We corrected the language errors as specified (now page 4 line 14; page 1 line 18; page 1 line 33). Thank you!

---

## Author Comment (AC4) · 20 Apr 2017

Thank you for your thoughtful comments. Including your suggested revisions has improved the quality of the manuscript. Our responses are indicated below in blue text.

This paper describes advancements in the modeling of global Hg cycling through up- dated redox chemistry as well as coupling to a more sophisticated ocean model. The approach is thorough and defensible and the paper well-written. I recommend publica- tion after minor revisions.

P5, l. 23-25: Can you add a summary of how accurate these OA concentrations are expected to be? If this isn't known, how does that impact your conclusions?

We respond to this and other reviewers' comments on a similar topic below.
We refer to a study evaluating modeled OA in detail, adding the following sentence (Page 6, lines 3-5): "An evaluation of modeled OA against aircraft observations globally is presented in Heald et al. (2011), which showed no systematic bias in remote environments but an underestimate of median concentrations in polluted regions."

P8, l. 1-20: This discussion of the mean and SD of TGM concentrations underplays the spatial deviations shown in Fig. 3. For example, the modeled and measured N-S gradients in Europe appear to be in opposite directions.
We reorder the discussion in this paragraph to highlight the spatial correlation coefficient (now page 8, lines 31-35).

P8, l. 28-30: I suggest you cite Soerenson et al., EST 2013, or other observation-based papers to show that this mechanism is consistent with observations.
We revise this section in response to this and others' comments (page 9, lines 22 to 24): "Modeled oceanic $Hg^0$ evasion from the North Atlantic Ocean peaks in winter and early spring, because higher windspeeds increase mixing and entrainment of reducible $Hg^{II}$ from subsurface waters and enhance the rate of air-sea exchange, and Hg removal via particle settling during this time is low (Sorensen et al., 2010, 2013)."

P9, l. 5-7: It sounds like you are suggesting that a similar analysis for the southern ocean is needed, but both the sea ice and the relatively large riverine flux relative to the Hg pool in the Arctic Ocean are not likely to have any relevance to these southern non-polar sites.

We clarify this section and include additional discussion (page 9, lines 35-37 to page 10, lines 1-3):
"Long-range transport of atmospheric Hg from Antarctica could influence observed seasonality at these sites (Angot et al., 2016). Capturing the seasonality of atmospheric Hg in the Arctic using GEOS-Chem required parameterization of unique sea-ice, oceanic, and riverine dynamics (Fisher et al., 2012; 2013; Y. Zhang et al., 2015) and a similar analysis for the Southern Ocean region, with its distinct productivity dynamics impacting Hg cycling (e.g., Cossa et al., 2011; Gionfriddo et al., 2016), has not yet been performed."

P10, l. 1-5: The figure indicates a -28% mean bias in North America, but in the Gulf of Mexico it appears to still be closer to -50%. Can you quantify the improvement in the wet deposition relative to Holmes et al. (or other models)? Consider adding a figure showing scatterplots of

model vs. observations for both TGM and wet deposition.

We rewrite this paragraph (now page page 10, lines 34-36 to page 11, lines 1-2) to address your and other reviewers' comments:

"Previous GEOS-Chem simulations with Br-initiated oxidation of $Hg^0$ failed to capture this maximum because $Hg^{II}$ production favored higher latitudes (Holmes et al., 2010; Amos et al., 2012). The inclusion of $NO_2$ and $HO_2$ as second-stage oxidants in our simulation shifts $Hg^{II}$ production to lower latitudes and matches the general location of the Gulf of Mexico maximum although the magnitude is still underestimated. Increasing horizontal resolution in the model could improve the definition of the maximum (Zhang et al., 2012)."

---

## Author Comment (AC5) · 20 Apr 2017

**A new mechanism for atmospheric mercury redox chemistry: Implications for the global mercury budget**

Hannah M. Horowitz[1], Daniel J. Jacob[1,2], Yanxu Zhang[2], Theodore S. Dibble[3], Franz Slemr[4],
5  Helen M. Amos[2], Johan A. Schmidt[2,5], Elizabeth S. Corbitt[1], Eloïse A. Marais[2], and Elsie M. Sunderland[2,6]

[1]Department of Earth & Planetary Sciences, Harvard University, Cambridge, MA, USA
[2]Harvard John A. Paulson School of Engineering and Applied Sciences, Harvard University, Cambridge, MA, USA
10  [3]Chemistry Department, State University of New York-Environmental Science and Forestry, Syracuse, NY, USA
[4]Max-Planck-Institute for Chemistry (MPI-C), Department of Atmospheric Chemistry, Mainz, Germany
[5]Department of Chemistry, University of Copenhagen, Universitetsparken 5, 2100 Copenhagen O, Denmark
[6]Department of Environmental Health, Harvard T. H. Chan School of Public Health, Boston, MA, USA

15  *Correspondence to:* Hannah M. Horowitz (hmhorow@post.harvard.edu)

**Abstract.** Mercury (Hg) is emitted to the atmosphere mainly as volatile elemental $Hg^0$. Oxidation to water-soluble $Hg^{II}$ plays a major role in Hg deposition to ecosystems. Here, we implement a new mechanism for atmospheric $Hg^0/Hg^{II}$ redox chemistry in the GEOS-Chem global model and examine the implications for the global atmospheric
20  Hg budget and deposition patterns. Our simulation includes a new coupling of GEOS-Chem to an ocean general circulation model (MITgcm), enabling a global 3-D representation of atmosphere-ocean $Hg^0/Hg^{II}$ cycling. We find that atomic bromine (Br) of marine organobromine origin is the main atmospheric $Hg^0$ oxidant, and that second-stage HgBr oxidation is mainly by the $NO_2$ and $HO_2$ radicals. The resulting chemical lifetime of tropospheric $Hg^0$ against oxidation is 2.7 months, shorter than in previous models. Fast $Hg^{II}$ atmospheric reduction must occur in order
25  to match the ~6-month lifetime of Hg against deposition implied by the observed atmospheric variability of total gaseous mercury (TGM $\equiv Hg^0 + Hg^{II}(g)$). We implement this reduction in GEOS-Chem as photolysis of aqueous-phase $Hg^{II}$-organic complexes in aerosols and clouds, resulting in a TGM lifetime of 5.2 months against deposition and matching both mean observed TGM and its variability. Model sensitivity analysis shows that the interhemispheric gradient of TGM, previously used to infer a longer Hg lifetime against deposition, is misleading
30  because southern hemisphere Hg mainly originates from oceanic emissions rather than transport from the northern hemisphere. The model reproduces the observed seasonal TGM variation at northern mid-latitudes (maximum in February, minimum in September) driven by chemistry and oceanic evasion, but does not reproduce the lack of seasonality observed at southern hemispheric marine sites. Aircraft observations in the lowermost stratosphere show a strong TGM-ozone relationship indicative of fast $Hg^0$ oxidation, but we show that this relationship provides only a
35  weak test of Hg chemistry because it is also influenced by mixing. The model reproduces observed Hg wet deposition fluxes over North America, Europe, and China with little bias (0-30%). It reproduces qualitatively the observed maximum in US deposition around the Gulf of Mexico, reflecting a combination of deep convection and availability of $NO_2$ and $HO_2$ radicals for second-stage HgBr oxidation. However, the magnitude of this maximum is underestimated. The relatively low observed Hg wet deposition over rural China is attributed to fast $Hg^{II}$ reduction
40  in the presence of high organic aerosol concentrations. We find that 80% of $Hg^{II}$ deposition is to the global oceans,

Hannah Horowitz 4/19/2017 1:45 PM

Hannah Horowitz 4/19/2017 1:45 PM

Hannah Horowitz 4/19/2017 1:45 PM

Hannah Horowitz 4/19/2017 1:45 PM

Hannah Horowitz 4/19/2017 1:45 PM

Hannah Horowitz 4/19/2017 1:45 PM

Hannah Horowitz 4/19/2017 1:45 PM

Hannah Horowitz 4/19/2017 1:45 PM

Hannah Horowitz 4/19/2017 1:45 PM

Hannah Horowitz 4/19/2017 1:45 PM

[revised manuscript text omitted]

---

## Author Comment (AC6) · 20 Apr 2017

[revised manuscript text omitted]

Hannah Horowitz 4/20/2017 2:12 PM
Hannah Horowitz 4/20/2017 2:12 PM
Hannah Horowitz 4/20/2017 2:12 PM
Hannah Horowitz 4/20/2017 2:12 PM

[Figure]

[Figure]

Hannah Horowitz 4/20/2017 2:12 PM

**Figure 5. Relationship between total gaseous mercury (TGM) and ozone concentrations in the lower stratosphere. TGM is shown as the decimal logarithm of the concentration in ng m$^{-3}$ STP. Observations are from CARIBIC commercial aircraft in the extratropical northern hemisphere for April 2014-January 2015 (Slemr et al., 2015). Tropospheric data as diagnosed by [O$_3$]/[CO] < 1.25 mol mol$^{-1}$ are excluded. Model points are data for individual flights sampled along the CARIBIC flight tracks on the GEOS-Chem 4°x5° model grid. Also shown are results from a model sensitivity simulation with a doubled Hg$^0$ oxidation rate in the stratosphere. Reduced-major-axis (RMA) regressions for the log(TGM)-ozone relationship are shown with slopes and correlation coefficients (r). Errors on the slopes are estimated by the bootstrap method.**

Hannah Horowitz 4/20/2017 2:12 PM

[Figure]

**Figure 6. Annual Hg wet deposition fluxes over North America, Europe, and China. Model values for 2009-2011 (background contours) are compared to 2007-2013 observations from the Mercury Deposition Network (MDN, National Atmospheric Deposition Program, http://nadp.sws.uiuc.edu/mdn/) over North America (58 sites), the European Monitoring and Evaluation Program (EMEP) over Europe (20 sites), and data from Fu et al. (2015, 2016) over China (9 sites). In the China panel, circles represent rural sites and diamonds represent urban sites as identified in Fu et al. (2015, 2016). For the MDN and EMEP networks, which collect weekly or monthly integrated samples, we only include sites with at least 75% of annual data for at least one year between 2007 and 2013; for China we only include sites with at least 9 months of data over the 2007-2013 period. MDN data are formally quality-controlled, while for EMEP data we rely on a subset of sites that have been quality-controlled (Oleg Travnikov, personal communication). Legends give the normalized mean bias (NMB) for all sites, excluding urban sites in China.**

[Figure]

**Figure 7. Annual 2009-2011 Hg$^{II}$ deposition fluxes in GEOS-Chem.**